# Germline Cas9 promoters with improved performance for homing gene drive

Jie Du[1] ✉, Weizhe Chen ®[1,2], Xihua Jia[1], Xuejiao Xu[1], Emily Yang[3], Ruizhi Zhou[1], Yuqi Zhang[1], Matt Metzloff ®[3], Philipp W. Messer ®[3] & Jackson Champer ®[1] ✉

Gene drive systems could be a viable strategy to prevent pathogen transmission or suppress vector populations by propagating drive alleles with super-Mendelian inheritance. CRISPR-based homing gene drives convert wild type alleles into drive alleles in heterozygotes with Cas9 and gRNA. It is thus desirable to identify Cas9 promoters that yield high drive conversion rates, minimize the formation rate of resistance alleles in both the germline and the early embryo, and limit somatic Cas9 expression. In *Drosophila*, the *nanos* promoter avoids leaky somatic expression, but at the cost of high embryo resistance from maternally deposited Cas9. To improve drive efficiency, we test eleven *Drosophila melanogaster* germline promoters. Some achieve higher drive conversion efficiency with minimal embryo resistance, but none completely avoid somatic expression. However, such somatic expression often does not carry detectable fitness costs for a rescue homing drive targeting a haplolethal gene, suggesting somatic drive conversion. Supporting a 4-gRNA suppression drive, one promoter leads to a low drive equilibrium frequency due to fitness costs from somatic expression, but the other outperforms *nanos*, resulting in successful suppression of the cage population. Overall, these Cas9 promoters hold advantages for homing drives in *Drosophila* species and may possess valuable homologs in other organisms.

Gene drive is a promising method to control pest insect populations and reduce the spread of vector-borne diseases. Engineered gene drives are designed to have higher inheritance rates than the normal 50% Mendelian expectation, allowing them to increase in frequency and eventually spread through a whole population[1,2]. Depending on the design goal, gene drives can be classified into two categories, modification and suppression. Modification drives could spread a desired cargo gene or another change into the target species' genome, while suppression drives are designed to reduce or eliminate the target species population for health, ecological, or economic purposes[3,4].

There are many types of gene drives, but CRISPR homing gene drive is the most widely studied and perhaps the most powerful. In heterozygotes with a homing drive allele, the wild-type allele can be converted into a drive allele (Fig. 1A) by homology-related repair (HDR). This process is called "drive conversion" or "homing." Biased inheritance occurs when germline cells are converted from drive heterozygotes to homozygotes. Alternatively, the wild-type allele could be converted into a resistance allele by end-joining repair, which often mutates the DNA's sequence, preventing recognition by the drive's guide RNA (gRNA)[1,5]. Ideally, Cas9 cleavage and HDR are confined to germline cells in early meiosis, which eventually form progeny.

However, drive conversion and resistance allele formation are not necessarily spatially restricted to germline cells. It can also occur due to Cas9 expression in somatic cells[6] (Fig. 1A) because gRNAs are usually ubiquitously expressed from U6 promoters. Temporally, such activity can also occur in germline precursor cells[7] and in zygotes or early

[1]Center for Bioinformatics, School of Life Sciences, Center for Life Sciences, Peking University, 100871 Beijing, China. [2]School of Life Sciences, Tsinghua University, 100084 Beijing, China. [3]Department of Computational Biology, Cornell University, Ithaca, NY 14853, USA. ✉e-mail: dujie123@pku.edu.cn; jchamper@pku.edu.cn

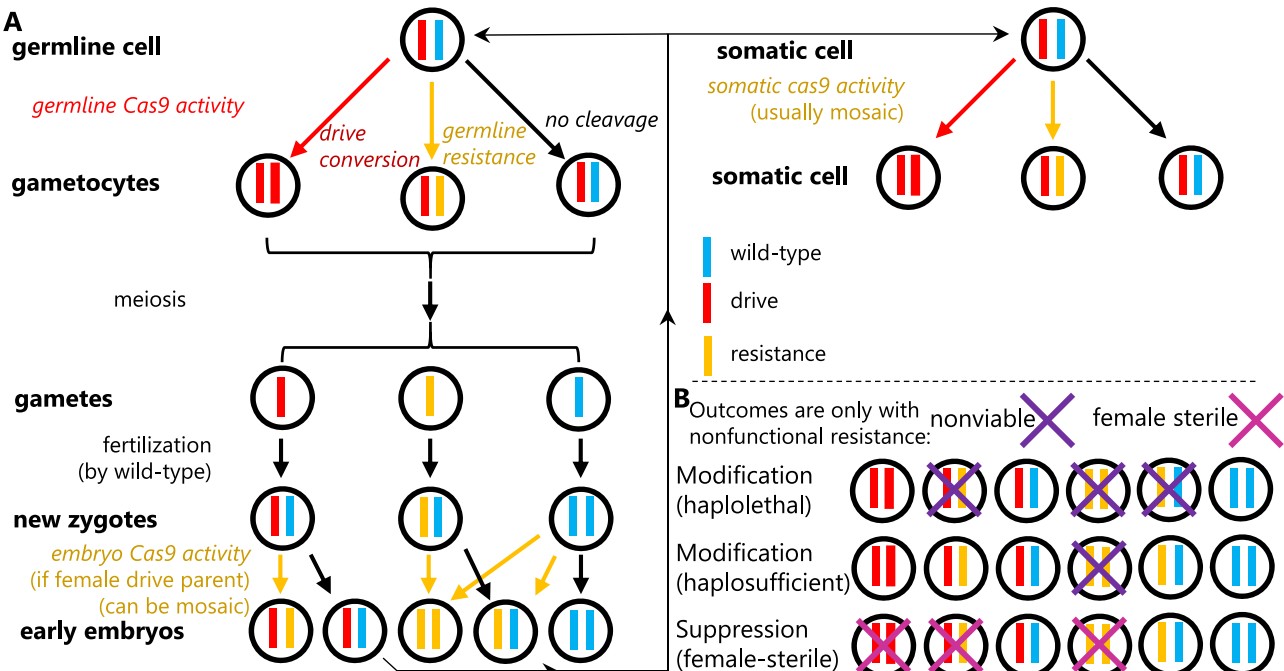

**Fig. 1 | Cas9 activity in homing gene drive. A** Drive conversion occurs in germline cells of drive/wild-type heterozygotes. Cas9 cleavage can result in wild-type alleles being converted into drive alleles by homology-directed repair, but resistance alleles can also be formed by end-joining. After meiosis and fertilization, maternal deposition of Cas9 and gRNA can form additional resistance alleles in the zygote or early embryo, a process that can be mosaic. Somatic Cas9/gRNA expression later in development or in adults can also result in drive conversion or resistance allele formation, though this process appears to be independent of germline activity. **B** Depending on the type of gene drive, certain individuals can be nonviable or sterile. In a rescue drive with a haplolethal gene target, any individuals with a nonfunctional resistance allele will be nonviable. With a haplosufficient target, only individuals with two nonfunctional resistance alleles are nonviable. In female-sterile suppression drive, only females must have at least one wild-type allele (or functional resistance allele) to be fertile. In general, functional resistance alleles have the same phenotype as wild-type alleles in all these drives. One exception is that they would not be susceptible to somatic expression and cleavage if together with a drive allele, potentially reducing fitness costs in suppression drives and haplolethal rescue drives.

embryos from parental Cas9 and gRNA deposition (Fig. 1A)[7–9]. Likely because of the larger relative size of female gametes, only maternal Cas9 deposition appears to occur regularly. Further, this process only forms resistance alleles rather than potentially supporting successful drive conversion, even in embryos that inherited a drive allele from the mother and a wild-type allele from the father[7–9]. Because embryo resistance and somatic expression often occur in only a fraction of cells, an individual could have a mosaic genotype due to variable Cas9 cleavage and repair outcomes in different cells. An ideal promoter for Cas9 results in a high drive conversion rate, low resistance allele formation rate, and low level of somatic expression.

Regardless of when they form, resistance alleles in drives with a specific target gene can be categorized as functional or nonfunctional, depending on whether they disrupt the function of the target gene. Such disruption can be from a frameshift mutation or other sufficient change in the protein's amino acid sequence. Functional resistance alleles tend to be less common because only one-third of indel mutations from end-joining will preserve the reading frame, and many of the remaining alleles will be nonfunctional due to changes in the target protein's amino acids. When the drive has a higher fitness cost than functional resistance alleles, the drive allele frequency will tend to be reduced over time. Fortunately, functional resistance can often be avoided by using multiplexed gRNAs[7,10] and conserved target sites[11]. Nonfunctional resistance alleles usually cannot outcompete a drive but can reduce its overall efficiency (see below).

Successful construction of homing drives has been achieved in many species, including yeast[12,13], mice[14], the fruit fly *Drosophila melanogaster*[15,16], and the mosquito species *Anopheles gambiae*[11], *Anopheles stephensi*[17], and *Aedes aegypti*[18]. Some homing endonuclease genes (HEGs) containing specific enzyme cut sites have been tested,

such as I-PpoI in *Anopheles*, but CRISPR/Cas9 is more flexible because its target sequence is determined by gRNA(s) rather than the nuclease itself[19]. Cas9-based drive efficiency tends to be quite high in yeast and *Anopheles* mosquitoes but lower in most designs for *Aedes*, flies, and especially mice. Homing gene drives can take many forms[3]. In the most basic form, they are unconfined to any target population and would spread widely, but variants such as split drive systems[20] and daisy chains[21,22] that separate Cas9 and gRNA elements can make them self-limiting. They would then be eliminated from the population after initially spreading under at least some parameter regimes. Confinement to target populations can also be achieved by targeting population-specific alleles[23] or by using tethered systems, where a confined type of drive provides the Cas9 for homing drives[24,25].

Aside from these variants, homing drives can be configured for either modification or suppression. Modification drives usually contain a cargo gene, exemplified by a drive in *A. stephensi*, where antipathogen effector genes targeting malaria parasites were successfully expressed[9]. Use of the *vasa* promoter here caused high rates of embryo resistance, but this was mitigated in an *A. gambiae* homing drive that used the *nanos* promoter for Cas9[26]. In *A. aegypti*, the *exu* and *nup50* promoters for Cas9 did not show high efficiency[27], and low drive conversion was also found with *sds3* and *bgcn*[28]. Despite working well in *Anopheles*, the *nanos* and *zpg* promoters also did not achieve drive inheritance rates above 75%[29]. When tested with a 4-gRNA construct, though, some genomic insertion sites for split Cas9 lines using the *shu* and *sds3* promoters showed high drive efficiency[18]. Somatic expression appeared to be moderate to high in all these *A. aegypti* lines. The most effective modification drives usually contain a recoded rescue element for an essential target gene, allowing the removal of nonfunctional resistance alleles. Targeting of a haplolethal gene

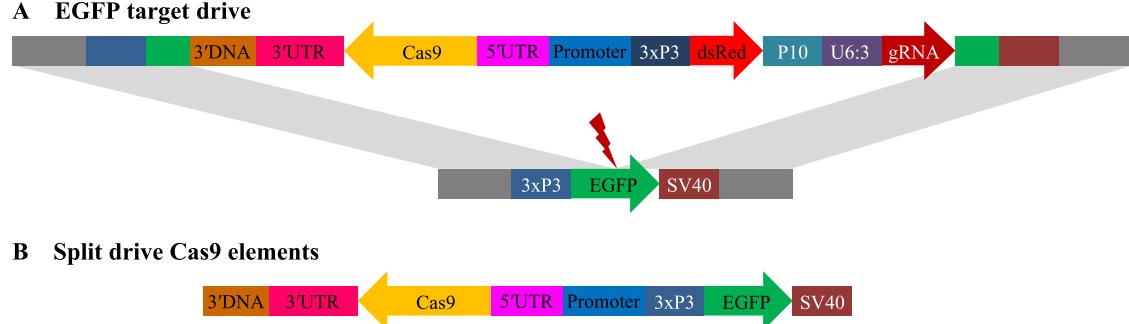

**Fig. 2 | Schematic diagram of main constructs in the study. A** The synthetic target drive is placed inside an EGFP gene at the gRNA target site. A DsRed fluorescence marker is regulated by the 3xP3 promoter for expression in the eyes together and a P10 3′ UTR element. A single gRNA driven by U6:3 promoter targets EGFP. Cas9 is driven by different compositions of promoter/5′ UTR and 3′ UTR. **B** The split drive Cas9 elements all contain an EGFP fluorescent marker gene driven by the 3xP3 promoter and with an SV40 3′ UTR. Cas9 is driven by different compositions of promoter/5′ UTR and 3′ UTR.

(where two functioning copies are required for viability) will allow immediate removal of nonfunctional resistance alleles (Fig. 1B), but embryo resistance can also remove drive alleles, and somatic expression can potentially form enough problematic nonfunctional resistance alleles that can reduce fitness. In a *D. melanogaster* example, embryo resistance was low enough to allow drive success[16], though this could potentially be an issue in other systems. When the target gene is haplosufficient[30] (a single wild-type or recoded drive copy is enough for viability), the only nonviable genotype for this is nonfunctional resistance allele homozygotes (Fig. 1B). Nonfunctional resistance allele removal is slower, but effects from embryo resistance from maternal deposition and somatic Cas9 cleavage will only slow the drive rather than potentially have substantial effects. This type of rescue was demonstrated in *A. stephensi*[17].

Suppression drives typically target essential but haplosufficient genes without providing a rescue[31]. By targeting female-specific genes, higher suppressive power can be achieved because drive alleles are only removed in sterile females that lack wild-type alleles (Fig. 1B) rather than in both sexes. Such drives eventually form homozygotes, which are nonviable or sterile, thereby removing drive alleles from the population. If the drive frequency reaches a high enough level, this can lead to population suppression but drives that lack sufficient genetic load will instead reach an equilibrium frequency with the population persisting (genetic load refers to the level of reduction in the reproductive potential of the population at this equilibrium). In *Anopheles gambiae*, three female fertility genes were selected for constructing gene drive systems[32]. However, aside from functional resistance, these drives suffered from high levels of embryo resistance from strong maternal deposition due to their use of the *vasa2* promoter for Cas9. Additionally, somatic Cas9 expression (together with gRNAs, which have thus far always been ubiquitously expressed) can render female drive/wild-type heterozygotes mostly sterile. Both of these factors reduce the genetic load of a suppression drive, though this reduction is large only when the drive does not have exceptionally high drive conversion. To reduce somatic expression and embryo resistance, *zpg*, *nanos*, and *exu* promoters were tested in *A. gambiae*, inspired by homology to known germline *Drosophila* genes[33]. The *exu* promoter showed low cut rates, but *zpg* and *nanos* had similar drive conversion to *vasa2* with much less embryo resistance and less somatic expression as well. Together with a conserved target site to avoid functional resistance, a homing suppression drive with the *zpg* promoter was able to eliminate an *Anopheles* cage population[11].

In *Drosophila*, the oldest HEG-based homing gene drives tested used a wide variety of 3′ UTRs and promoters, including *β-Tub85D*, *Mst87F*, *Hsp70Ab*, *vasa*, *Act5C-P*, *aly*, *bgcn*, *rcd-1r*, and *CG9576* for different nucleases[34–37]. None of these achieved high efficiency, though *rcd-1r*, *hsp70Ab*, and *Act5C-P* were able to promote some drive conversion. Cas9-based systems using *vasa* performed better, albeit with high rates of embryo resistance and somatic expression[7]. The *nanos* promoter had a similar performance without apparent somatic expression[7,8]. The *rcd-1r* promoter was also tested at two target sites with similar performance, though only drive conversion was evaluated[38,39]. Some of these promoters, together with *exu*, were evaluated in another recent study, but these Cas9 genes used a T2A fusion to EGFP, as well as the P10 terminator element. Either of these may substantially change gene expression patterns. Some achieved high drive conversion efficiency based on the gRNA target site, but embryo resistance and somatic expression were not evaluated. Thus, despite being a model organism, Cas9 promoters in *D. melanogaster* have achieved less efficiency than *Anopheles* and perhaps even *Aedes*. This is showcased by a *nanos*-Cas9 suppression drive experiment that avoided functional resistance alleles but failed due to inadequate drive conversion efficiency, high embryo resistance, and high fitness costs[15].

To improve *Drosophila melanogaster* homing gene drive efficiency in this study, we constructed and tested eleven germline Cas9 promoters in different configurations. Some promoters resulted in a higher drive conversion rate and lower embryo resistance rate. However, unlike the *nanos* promoter, none were able to avoid somatic expression. Furthermore, three Cas9 constructs were selected for cage experiments with a 4-gRNA suppression drive. One of these had significantly better performance than *nanos*[15], resulting in the successful elimination of the cage population. Our results demonstrate that these Cas9 promoters could be useful in *Drosophila* homing gene drive systems.

## Results
### Cas9 regulatory element selection and construction
In this study, we constructed two types of drive systems. In our synthetic target drives, the homing drives are complete, and target EGFP (Fig. 2A) is placed at "site C" on chromosome 2L[10]. The other system uses split Cas9 elements (Figs. 2B and S1A) that are usually placed at "site B" on chromosome 2R[20]. These are then paired with one of three possible split drive elements for drive efficiency assessment. Each of these Cas9 elements insertion sites is downstream of two genes on either side (adjacent to the 3′ UTR of both genes) to minimize fitness costs or other interference between genes. All drive elements have DsRed fluorescent markers, while Cas9 elements are marked with EGFP.

Two of our drive systems are designed for easy visualization of nonfunctional resistance alleles, including the EGFP target drives and a split driving element targeting the X-linked *yellow* gene. Phenotypes for the EGFP drive are shown in Fig. S1B, and functional resistance alleles are rare for this drive despite it having just one gRNA. For the X-linked drive targeting *yellow*[20], null alleles have a recessive yellow body color phenotype (these can be driven or nonfunctional resistance

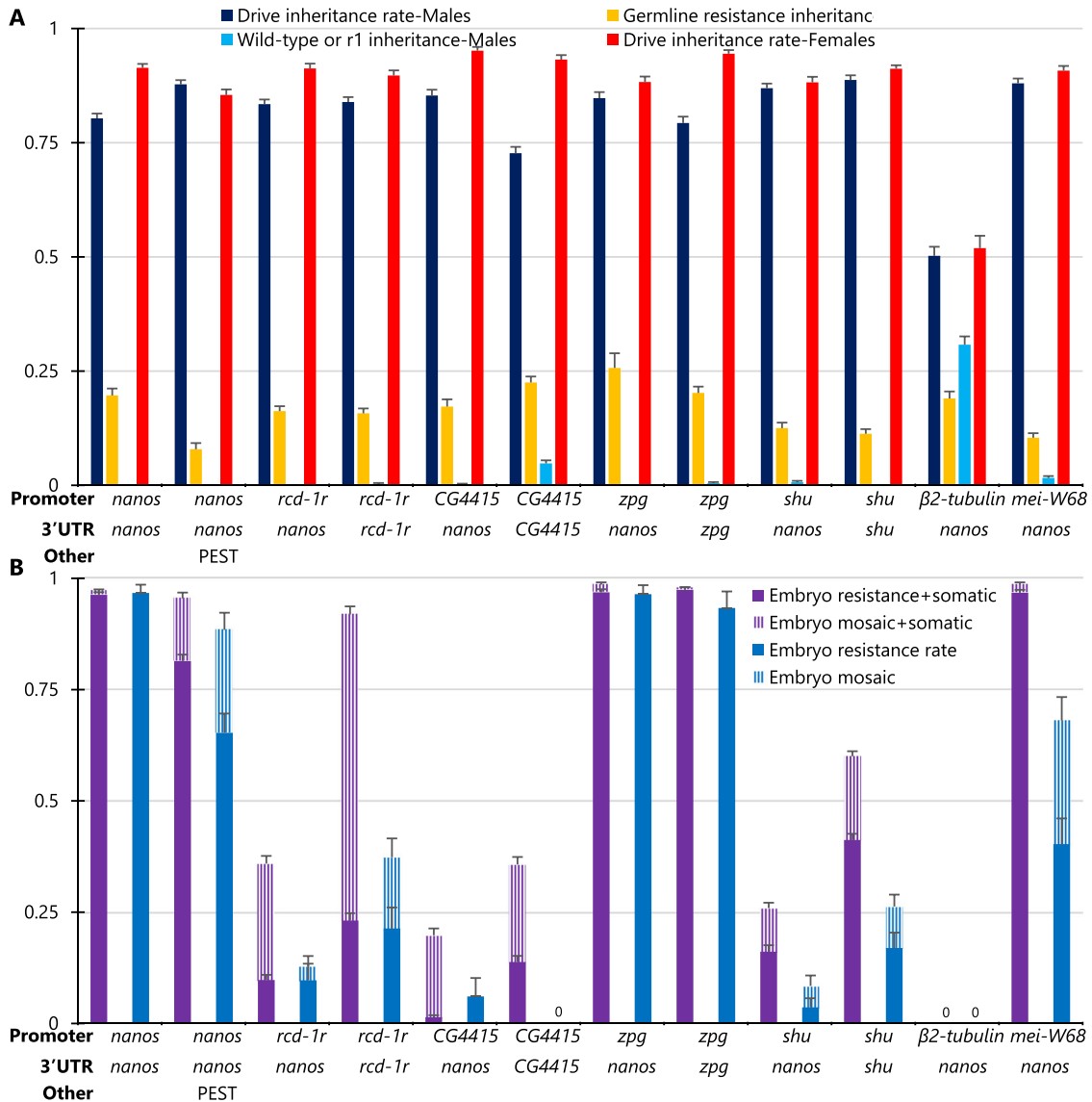

**Fig. 3 | EGFP target site drive performance.** The chart shows the drive performance of twelve homing drive systems targeting EGFP on chromosome 2L differing by promoter/5′ UTR and 3′ UTR regulation of Cas9, or in one case, addition of a PEST sequence inside Cas9. **A** Drive inheritance for males and females was measured in the progeny of drive/EGFP heterozygotes. Female germline resistance was not measured, and male germline resistance and wild-type inheritance were measured from crosses with w[1118] females (some male crosses were with EGFP homozygous females, and these crosses only contribute to driving inheritance measurements). Female drive heterozygotes were always crossed with EGFP males. **B** The fraction of offspring lacking EGFP phenotype (or with mosaic phenotype) and inheriting the drive is labeled as "Embryo resistance+somatic" because either maternally deposited Cas9 and gRNA or somatic expression in the eye could be responsible for lack of EGFP. "Embryo resistance rate" (and the corresponding mosaic rate) is similar but reports the fraction of non-drive offspring lacking EGFP, which can only be caused by maternal deposition. The leftmost drive data is from a previous study[10]. Error bars represent SEM. Source data is provided in Data Set S1 in the Source Data file.

alleles, see Fig. S1C), and functional resistance allele represents -10% of total resistance alleles[8,20]. With our split Cas9 elements, we also tested a haplolethal drive targeting *RpL35A* with two gRNAs[16] (Fig. S1D) that have several nonviable genotypes and also a suppression drive targeting the haplosufficient, female fertility *yellow-G* gene with four gRNAs[15], which has several genotypes that are female sterile or reduced fertility (Fig. S1E).

A list of all constructs used in the study can be found in Table S1, and Table S2 contains details of the sizes of our regulatory elements, including the promoter (defined here as DNA before the 5′ UTR), 5′ UTR, 3′ UTR, and included DNA downstream of the 3′ UTR. In general, the entire 3′ UTRs were used. We also added a small amount of additional DNA beyond the 3′ UTRs in case this was important for transcription termination. For promoters, we usually used DNA that did not overlap with other genes or an area immediately upstream of the 5′

UTR of other genes, which likely contained a core promoter of that gene. In many cases, though, this would have resulted in a very small promoter. In such cases, we often included 3′ UTRs of other genes or even some of the 5′ UTRs. For *nanos* and *vasa*, we used existing constructs as a basis[8]. Other promoters were selected for germline-restricted expression and low mRNA levels in the early embryo, according to the Berkeley *Drosophila* Genome Project (https://insitu.fruitfly.org)[40]. *zpg* was selected due to its high efficiency in an *Anopheles* homing drive[11], and *β2-tubulin* was selected because it is a known male-restricted germline promoter[41,42].

## Comparative drive performance at an EGFP target site
We designed and tested 12 enhanced green fluorescent protein (EGFP) target drives composed of different promoter and 3′ UTR elements in *D. melanogaster* (Fig. 2A). These are similar to a drive

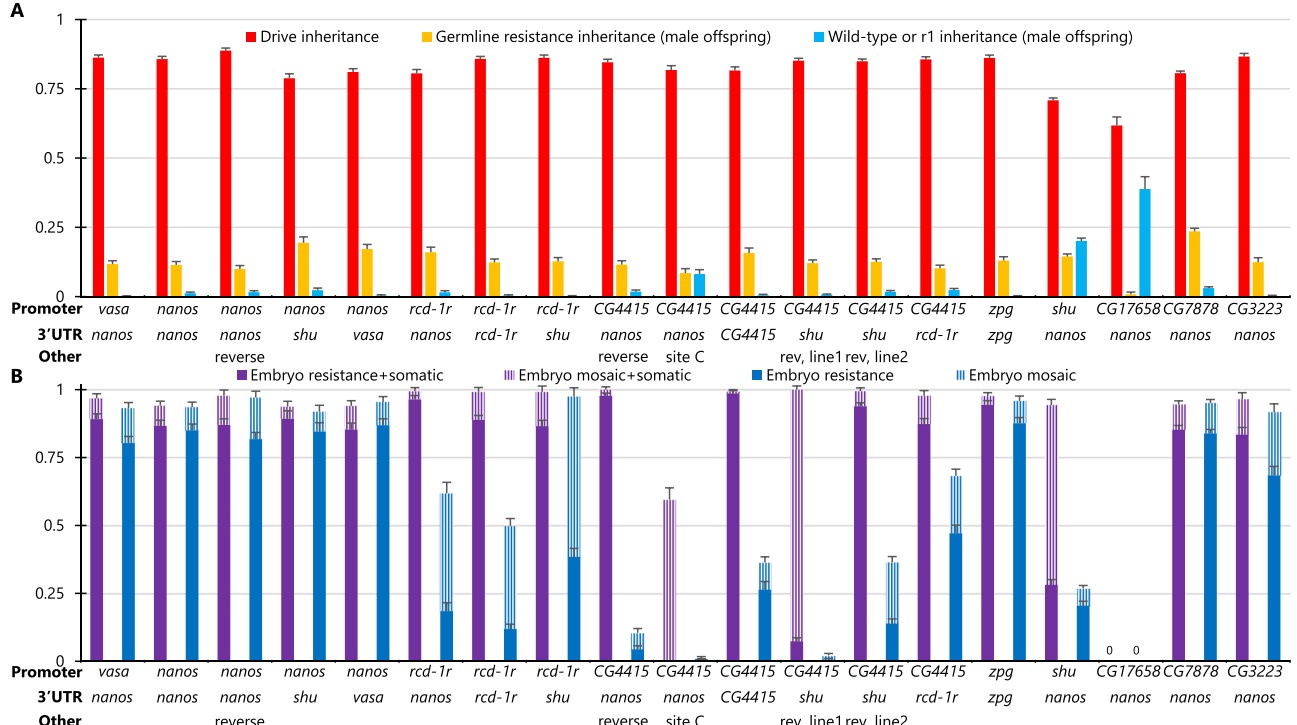

**Fig. 4 | *yellow* target site drive performance.** Females heterozygous for different Cas9 alleles and for the *yellow* drive were crossed with $w^{1118}$ males. **A** Their progeny were phenotyped for DsRed (drive), EGFP (Cas9), and yellow body color. The germline resistance inheritance shows the fraction of male progeny with yellow body but no drive, and wild-type and r1/functional resistance indicates the fraction of male offspring that were wild-type. **B** The fraction of offspring with yellow (or mosaic) phenotype inheriting the drive and also inheriting Cas9 is "Embryo resistance+somatic" because either maternally deposited Cas9/gRNA or somatic expression could cause the yellow phenotype. "Embryo resistance rate" (mosaic rate) reports the fraction of drive offspring lacking Cas9 that have the yellow phenotype, which must be from maternal deposition. "Reverse" indicates a change in orientation on one gene of the allele so that the Cas9 and 3xP3 promoters are not adjacent. One Cas9 allele had different performances between lines, displayed as "line 1" and "line 2". Error bars represent SEM. Source data is provided in Data Set S2.

described previously[10] that used the *nanos* promoter, 5′ UTR, and 3′ UTR. Though the drive uses only one gRNA, nearly all resistance alleles are nonfunctional, which disrupts EGFP and allows for the determination of most drive performance parameters without sequencing (Fig. S1B).

To determine drive performance, offspring from drive/EGFP drive heterozygotes were phenotyped (Fig. S2). In particular, female virgin drive/EGFP heterozygotes are crossed with males homozygous for EGFP. Drive/EGFP heterozygote males were crossed to EGFP homozygous or $w^{1118}$ female virgin flies (Fig. S2). Note that rather than reporting the standard parameter of drive conversion efficiency (the percentage of EGFP alleles converted into drive alleles in germline cells) and resistance allele formation rate, we report inheritance rates for compatibility with our haplolethal-targeting drive (see below). This is because in this drive, offspring with nonfunctional resistance alleles are nonviable, so the drive conversion rate cannot be calculated based only on the drive inheritance.

Most drive systems showed 72–89% drive inheritance rates for males and 85–95% for females (significantly different from the Mendelian expectation, $P < 0.0001$ binomial exact test), with females having consistently slightly better performance (except for the one with the PEST sequence, see below) (Fig. 3A, Source Data–Data Set S1). However, one drive system with the *β2-tubulin* promoter showed only Mendelian inheritance for both males and females. Even though *β2-tubulin* did not show any drive conversion, embryo resistance, or somatic activity, it still had some germline resistance formation in males. For all other promoters, the total germline cut rate (drive conversion plus germline resistance allele formation) was usually 100% as measured in crosses between drive males and $w^{1118}$ females. Drive inheritance rates for other constructs were generally similar. The drive with the *shu* promoter and 3′ UTR had the highest drive inheritance rate of almost 89% for males, and the drive with the *CG4415* promoter and *nanos* 3′ UTR had the highest drive inheritance rate in females of 95%. Only the drive with the *CG4415* promoter and 3′ UTR in males had a notably lower inheritance rate of 72%. Because these very different promoters showed similar germline performance, it is possible that Cas9 cut rates were highly saturated in the germline, perhaps due to the use of a high-activity gRNA. However, absolute germline expression levels or at least timing likely still varied greatly based on their highly different embryo resistance allele formation rates, though this is an indirect proxy.

Patterns in the embryo resistance rate in the progeny of females varied more substantially between drive lines (Fig. 3B, Source Data–Data Set S1). This can only be directly measured in flies lacking a drive allele because somatic expression can also remove the EGFP phenotype or cause mosaicism. Except for the *nanos* and *β2-tubulin* promoters, all tested promoters showed moderate to high levels of somatic expression, resulting in mosaic drive/EGFP heterozygous parents. The *nanos* promoter had only low levels of eye mosaicism, perhaps from proximity to the 3xP3 promoter, and the *β2-tubulin* promoter had minimal expression in general. We also saw the fluorescent expression in the gonads of males and females with *nanos* adjacent to 3xP3, and in males with *rcd-1r*, indicating that 3xP3 at least can be affected by adjacent enhancer elements. To account for this possible leaky expression of Cas9 in the eyes from 3xP3, the mosaic phenotype was thus scored only for individuals that had at least 1/3 absence of EGFP in the surface of at least one eye. This was found to be sufficient to avoid scoring most individuals as mosaic when this was

caused entirely due to somatic Cas9 expression, except for the *mei-W68* promoter with its very high somatic expression. However, because germline cut rates were generally 100% in females, the embryo resistance allele formation rate could be measured directly in individuals who failed to inherit the drive. This is because they would still almost certainly inherit a nonfunctional resistance allele from the mother, requiring additional cleavage only in the paternal EGFP allele. Some drive systems showed very high embryo resistance rates, such as those with the *nanos* and *zpg* promoters and, to a lesser extent the *mei-W68* promoter. The drives with *rcd-1r*, *shu*, and particularly the *CG4415* promoter showed much lower embryo resistance. For these, the use of the *nanos* 3′ UTR tended to give slightly lower embryo resistance than the corresponding 3′ UTR of the promoters.

In the majority of flies inheriting the drive, lack of EGFP could be caused by either somatic expression or embryo resistance, but if the embryo resistance allele formation rate was high, then additional somatic expression would have little effect. Nevertheless, we saw notable increases in progeny that lacked EGFP phenotype or were mosaic in individuals inheriting drives with *rcd-1r*, *CG4415*, *shu*, and *mei-W68* promoters, which was less when the *nanos* 3′ UTR was used (Fig. 3B, Source Data–Data Set S1). Overall, the *rcd-1r*, *CG4415*, and *shu* promoters appeared to be promising combinations with the *nanos* 3′ UTR for high drive performance. These still had more somatic expression than the *nanos* promoter, but it was kept to a moderate level, and they had very low embryo resistance allele formation rates.

### Split drive performance at the *yellow* gene

Our EGFP target drives allowed an initial assessment of promoter performance in males and females, but they did have some disadvantages. First, they could only detect cutting activity in the eyes, but important somatic expression may be present in other tissues. Second, they made it difficult to distinguish between somatic expression and embryo resistance because most offspring inherited the drive, resulting in low sample sizes for the calculation of embryo resistance. Third, they weren't compatible with several newer split driving elements that were specialized for modification and suppression, representing drives closer to field applications. Thus, we designed and constructed several split Cas9 elements, mostly at the same genomic locus. One with the *CG4415* promoter was placed at "site C", where EGFP target drives were shown to have higher drive conversion compared to our default "site B" locus[10,20]. We first combined these Cas9 elements with a split drive targeting *yellow* (Fig. S1C), which tends to have somewhat lower embryo resistance than the EGFP drives[20]. It is also X-linked, allowing assessment of germline resistance inheritance from females (male offspring will only have one copy of *yellow* from their mother). Recessive knockout alleles cause a whole-body phenotype, allowing a different assessment of somatic expression. However, only drive performance in females can be tested.

Drive assessment was conducted by first crossing males homozygous for the Cas9 element to females that were homozygous for the drive element (Fig. S2). Then, drive/Cas9 heterozygous female virgins were crossed with *w[1118]* males. DsRed fluorescence for the drive element, EGFP fluorescence for the Cas9 element, and yellow body color phenotype were scored to assess drive performance (Fig. 4A, Source Data–Data Set S2). In 17 of the 19 Cas9 elements with varying promoter and other factors, the drive inheritance rate mostly ranged from 79% to 89% (significantly different from the Mendelian expectation, $P < 0.0001$ binomial exact test). The total apparent cut rate (drive conversion plus nonfunctional germline resistance allele formation) for these was usually very close to 100%. Because functional resistance alleles appear as wild-type, the actual cut rate was likely 100% in many cases, considering the relatively high functional resistance allele fraction at this target site[8]. The *shu* and *CG17658* promoters had lower

drive inheritance rates of 71% and 62%, respectively, and had total cut rates significantly below 100%. The *CG4415* promoter drive at "site C" also did not achieve complete germline cutting. The *nanos* promoter and 3′ UTR showed the highest drive inheritance of 88.7%.

Only three Cas9 promoter elements with high drive inheritance rates avoided high embryo resistance (Fig. 4B, Source Data–Data Set S2). These were *rcd-1r*, *CG4415*, and *shu*, though our test with *shu* showed less efficiency for drive inheritance. Of these, *CG4415* with either the *nanos* or *shu* 3′ UTR (but not the *CG4415* 3′ UTR) had the lowest embryo resistance (1-4%).

Somatic Cas9 expression was more prevalent for the *yellow* split drive than the EGFP target drives. In the initial cross, only drive heterozygous females with the *nanos* and *CG17658* promoters (the latter of which had very little activity in general) had no sign of any yellow mosaicism in any flies, which would be indicative of somatic expression. For the *vasa*, *zpg*, *CG7878*, and *CG3223* promoters, somatic expression was always moderate to high. For other lines, the level of somatic expression can be quantitatively assessed by comparing female progeny with and without the Cas9 allele. Both can have embryo resistance alleles, but somatic Cas9 expression can only occur in progeny with a Cas9 allele. This allowed us to see moderate somatic expression in most remaining Cas9 lines based on the *rcd-1r*, *CG4415*, and *shu* promoters. In three lines based on the *nanos* and *CG4415* promoters, we reversed the orientation of the EGFP and Cas9 promoters to prevent the 3xP3 of the EGFP marker from potentially inducing somatic expression of Cas9. However, this was not necessary to avoid visible somatic expression with *nanos* for the *yellow*-targeting drive, and somatic expression remained in the *CG4415* lines (Fig. 4B, Source Data–Data Set S2). A more effective strategy involved placing Cas9 with the *CG4415* promoter at "site C," which reduced somatic expression.

When assessing drive performance for these Cas9 elements, different lines were obtained from the same original injection. These usually showed the same performance and were thus combined in our analysis (a small number of lines showed no drive activity and were discarded). However, two sublines with the *CG4415* promoter and *shu* 3′ UTR showed significantly different performance. The second line had notably higher embryo resistance and somatic expression ($P < 0.0001$ Fisher's exact test). Genotyping detected no apparent difference in the insertion site, promoter/5′ UTR sequence, 3′ UTR sequence, or Cas9 itself, so it is unclear what caused the performance difference between these lines.

### Addition of a PEST domain for increased Cas9 degradation

One variant with the *nanos* promoter in our EGFP drives involved adding a PEST sequence to the C-terminus of Cas9. Such PEST sequences are known to increase the rate of protein degradation, and we hypothesized that this could reduce the level of effective maternal Cas9 deposition and thus reduce embryo resistance. Among progeny inheriting a drive allele, embryo resistance (somatic expression would not likely be a large factor in this *nanos* drive) was modestly reduced from 96% to 81%.

We thus decided to test several variants of split Cas9 elements with PEST sequences, some of which had reversed orientation between the Cas9 promoter and 3xP3 (Fig. S2). Unfortunately, these drastically reduced the drive inheritance rate (Fig. S3A, Source Data–Data Set S3). While embryo resistance rates and somatic expression levels were also reduced (Fig. S3B, Source Data–Data Set S3), the germline activity of this driving element was perhaps less highly saturated than the EGFP drives, resulting in the PEST addition, causing a large reduction in germline cut rates. Even the strong *vasa* and *CG7878* promoters had drive inheritance rates of under 70%, and several drives were statistically indistinguishable from the Mendelian expectation. All still had at least some germline cleavage activity, and all except those with the *nanos* promoter had some noticeable somatic activity.

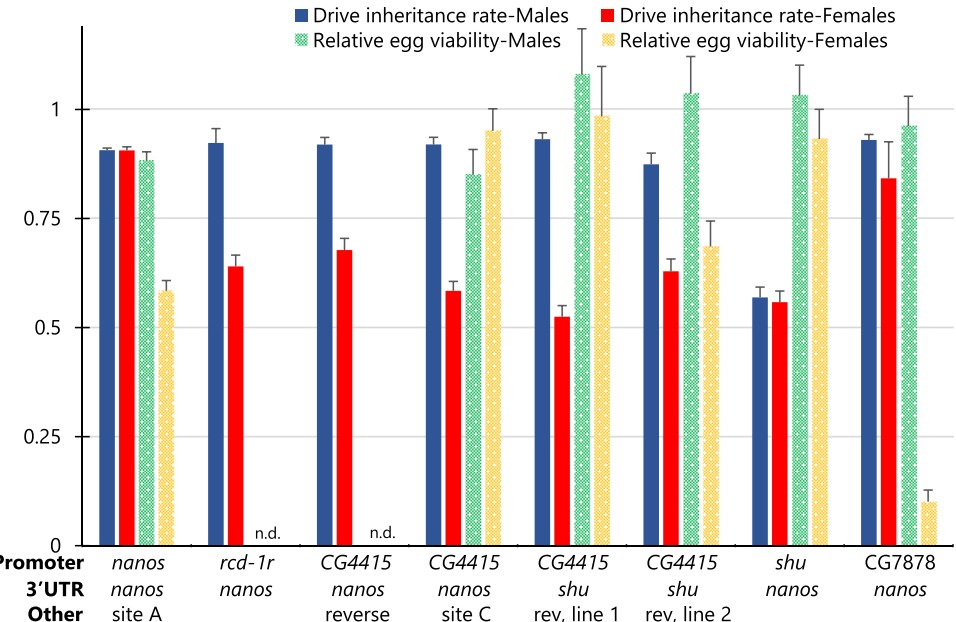

**Fig. 5 | Drive inheritance and offspring viability of promoters with haplolethal homing rescue drive.** Flies heterozygous for different Cas9 alleles on chromosome 2R (2L for "site C") and a drive targeting the haplolethal *RpL35A* gene were crossed with *w^{1118}* flies. *RpL35A* is a haplolethal gene, so progeny with resistance alleles were nonviable, and high somatic expression could also potentially reduce the viability of offspring that inherit both drive and Cas9. Progeny was phenotyped for DsRed (drive), and for several experiments, eggs were counted after one day of egg laying. Relative egg viability is the relative rate of egg survival compared to control experiments in which egg viability was measured for drive heterozygotes without Cas9, Cas9 heterozygotes, and crosses between *w^{1118}* flies. "Reverse/rev" indicates that the orientation on one gene of the allele is reversed so that the Cas9 promoter and 3xP3 of EGFP are not adjacent. n.d. not determined. The leftmost drive data is from a previous study and has a genomic site located 277 bases away from our default Cas9 insertion site on chromosome 2R[16]. Error bars represent SEM. Source data is provided in Data Set S4.

## Performance with a modification drive with haplolethal rescue

A homing rescue drive for population modification allows the removal of resistance alleles by targeting an essential gene. When the target is haplolethal, any nonfunctional resistance alleles cause nonviability. Thus, embryo resistance is harmful, but resistance alleles in general can be removed quickly. The effect of somatic expression is less clear. If it tends to result in drive conversion in somatic cells, fitness effects may not be large. However, it is also possible that even mild somatic expression would form enough nonfunctional resistance alleles to induce severe fitness costs. We assessed these possibilities by combining eight of our split Cas9 lines with good performance together with a previously constructed 2-gRNA haplolethal homing drive[16].

Drive homozygous females were crossed to Cas9 homozygous males, and the heterozygote progeny were then crossed to *w^{1118}* flies (Fig. S2). In some cases, flies were allowed to lay eggs for 20–24 h periods before being moved to each vial, and the eggs were counted to allow for assessment of egg viability. Except for a Cas9 element driven by the *shu* promoter, all tested promoters showed high drive inheritance rates for males, ranging from 87% to 93% (Fig. 5, Source Data–Data Set S4). However, only two Cas9 elements driven by the *nanos* and *CG7878* promoters had drive inheritance rates for females of over 70%. This is likely the result of reduced germline expression with these promoters, at least in females, coupled with reduced gRNA activity in this driving element compared to the split drive targeting *yellow* (best seen by comparing embryo cut rates in these systems[16,20]).

For egg viability experiments, several controls were used of the same age and often in the same vials as the drive/Cas9 flies. The relative egg viability was assessed compared to these controls. Of the six promoters that underwent egg viability assessment (Fig. 5, Source Data–Data Set S4), all had high relative egg viability in the progeny of males, ranging from 0.8 to 1.1 (values above 1 merely represent slightly higher viability than wild-type controls). In the progeny of males, nonviability can be the result of germline resistance alleles, which

likely occur at low frequency for this drive[16]. Fitness costs from a somatic expression could also potentially cause nonviability, and this could occur in drive-carrying offspring who also inherited a Cas9 allele. Nonviability due to somatic expression would also be expected to reduce the frequency of Cas9 inheritance in the progeny of drive heterozygous males. This was not observed (Source Data–Data Set S4), indicating that even when somatic expression is moderate (as in the *CG7878* promoter line), fitness costs from somatic expression are low. In the female lines, embryo resistance is also a factor. It is low for most of these lines, so the viability of the progeny of female drive individuals was also usually high. However, the *CG7878* promoter resulted in offspring viability lower than 0.1, most likely due to higher embryo resistance.

## Suppression drives performance in individual crosses

Homing suppression drives targeting haplosufficient but essential female fertility genes have the potential to induce strong suppression with good performance parameters. However, if drive conversion is not very high, then embryo resistance and fitness costs in heterozygous females can reduce the suppressive power of the drive by reducing its equilibrium frequency. This was the case with our 4-gRNA drive targeting *yellow-G*[15], an eggshell protein that is critical for egg development. Fitness costs due to somatic Cas9 expression can have a severe effect. Unlike in modification drives, drive conversion in somatic cells would also disrupt the target gene and reduce fitness. For this drive, however, fitness costs were observed even with the *nanos* promoter, indicating that disruption of at least this specific target gene in the germline also reduces fertility. None of our promoters had less somatic expression than *nanos*, but they often had less germline expression, potentially reducing fitness costs, and several had lower embryo resistance.

With a similar experimental setup to our investigation of the haplolethal split homing drive (Fig. S2), we found that drive

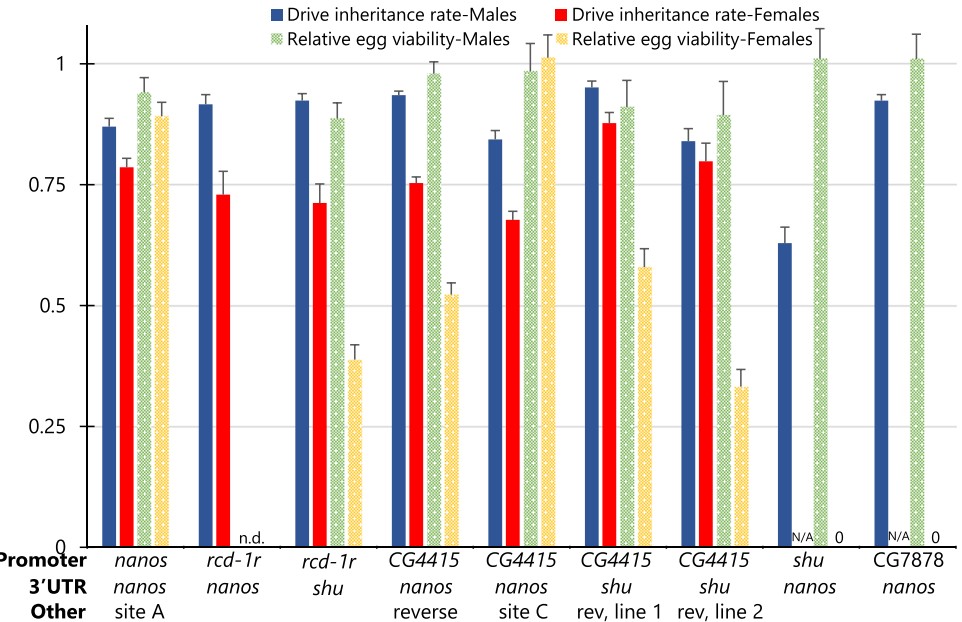

**Fig. 6 | Drive inheritance and offspring viability of promoters with homing suppression drive.** Flies heterozygous for different Cas9 alleles on chromosome 2R (2L for "site C") and a drive targeting *yellow-G* gene on chromosome 3 were crossed with *w1118* flies. *yellow-G* is a haplosufficient gene essential for female fertility, so progeny from females suffering from high somatic Cas9 expression (or potentially high germline expression) have lower viability. Progeny was phenotyped for DsRed (drive), and for several experiments, eggs were counted after one day of egg laying. Relative egg viability is the relative rate of egg survival compared to control experiments in which egg viability was measured for drive heterozygotes without Cas9, Cas9 heterozygotes, and crosses between *w1118* flies. "Reverse" indicates that the orientation on one gene of the allele is reversed so that the Cas9 promoter and 3xP3 of EGFP are not adjacent. n.d.−not determined, N/A−not applicable (no offspring to measure inheritance), 0−no viable eggs. The leftmost drive data is from a previous study[15]. Error bars represent SEM. Source data is provided in Data Set S6.

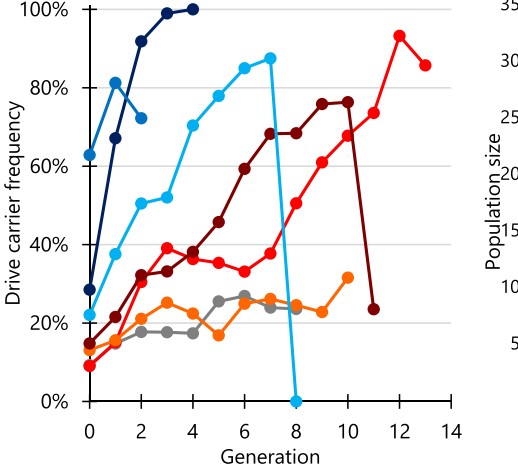
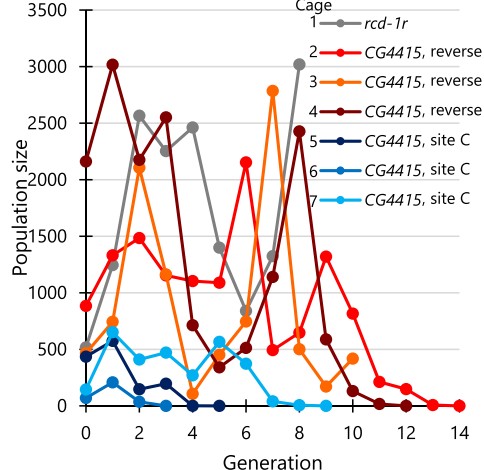

**Fig. 7 | Multigenerational cage experiments with homing suppression drive.** Cage experiments were initialized in generation zero, which were progeny of Cas9 homozygous females mated with either Cas9 homozygous males or drive heterozygous males (that were also homozygous for Cas9). Cas9 alleles were either *rcd-1r* line #1 with the *shu* 3′ UTR, *CG4415* with the *nanos* 3′ UTR in reverse orientation, or *CG4415* with the *nanos* 3′ UTR at site C. The cage populations were maintained separately with nonoverlapping generations, each lasting 12−13 days with 1 day for egg laying. All individuals for each generation were phenotyped for DsRed (indicating drive carriers that could be homozygous or heterozygous), and the total population was also recorded. Five of the seven cages resulted in population elimination (all except the *rcd-1r* cage and the *CG4415* cage in orange). Note that in two cages, the drive carrier frequency fell substantially in the last generation because only a small number of flies were left (because population elimination occurred, individuals that were not drive carriers were likely all males or sterile resistance allele homozygotes). Source data is provided in Data Set S6.

inheritance from males was also consistently higher than from females (Fig. 6, Source Data−Data Set S5), though differences were smaller than for the drive targeting *RpL35A*. Except for the *shu* promoter, which had worse performance, all other promoters had high drive inheritance rates for males ranging from 84% to 95%. In the progeny of females, drive inheritance ranged from 67% to 87%. Eggs from male parents retained high viability (between 0.89 and 1.04). However, most females had lower egg viability (Fig. 6, Source Data−Data Set S5). While the *nanos* promoter showed a small reduction, others had more substantial reductions, and somatic expression from the *shu* and *CG7878* promoters resulted in no eggs being viable. Only the *CG4415* promoter placed at chromosome 2L retained high egg viability. Though it has

more somatic expression than *nanos*, this increased fitness may come from less germline expression or perhaps a different spatial or temporal pattern of expression in the germline or ovaries in general. However, somatic expression remains important for fitness with this drive, as evidenced by the *shu* promoter, in which no eggs were viable despite apparently low germline cut rates.

For two drives, embryo resistance was inferred by measuring if female drive carrier progeny of females with the drive and Cas9 were fertile. Infertility would be caused if the wild-type copy of *yellow-G* that these progeny receive from their father is converted into a nonfunctional resistance allele in the early embryo. For the Cas9 with the *CG4415* promoter, *nanos* 3′ UTR, and reverse orientation, all 36 females tested were fertile (as well as 21 similar female controls that were the offspring of males with drive and Cas9). When the *rcd-1r* promoter was used with the *shu* 3′ UTR, all 13 females were fertile (as were all 23 control females as above). This indicates that the suppression drive targeting *yellow-G* likely had naturally lower embryo resistance than the drives targeting EGFP or *yellow*, allowing our promoters with lower embryo resistance in these other systems to also avoid high embryo resistance with the suppression drive.

### Suppression drive cage experiments

Previously, our 4-gRNA homing suppression drive targeting *yellow-G* failed to reduce the size of two cage populations, reaching only an intermediate equilibrium frequency[15]. The *nanos* promoter only showed small fitness costs in individual crosses, but it had substantially higher fitness costs in cage populations, perhaps due to different environmental conditions, such as increased desiccation risk. High embryo resistance in the *nanos* promoter also contributed to poor performance. We selected three split Cas9 lines for similar cage experiments. Two used the *CG4415* promoter and *nanos* 3′ UTR, one of which had the Cas9 in the same orientation as EGFP, and the other of which was placed at a different chromosome arm. The third used the *rcd-1r* promoter and *shu* 3′ UTR (line #1 for higher performance).

First, Cas9 homozygous female virgins were collected, mixed, and then mated to either drive heterozygous males or males that were wild-type at the drive site (all males were also homozygous for Cas9). Then, males were removed, and females were allowed to lay eggs in cage bottles for two days. Females were removed, and new food was provided to offspring eleven days later. These offspring were considered to be "generation zero," in which the drive heterozygote frequency was approximately 10%. Flies were then kept on a 12-day cycle with discrete generations and approximately 24 h of egg-laying per generation. All flies were phenotyped to track the drive carrier frequency and total population size.

In the cage with Cas9 driven by the *rcd-1r* promoter, drive carrier frequency slowly increased at first but always remained lower than 27%, apparently reaching a low equilibrium value (Fig. 7, Source Data—Data Set S6). The total population size was not affected, fluctuating from a maximum of 3019 adults to a minimum of 840.

For cage 2 with Cas9 element driven by *CG4415* promoter (Fig. 7, Source Data—Data Set S6), the drive carrier frequency increased quickly, then remained constant for a few generations. With controlled temperature and humidity, this sudden change in behavior was possibly due to random fluctuations or, more likely, differences in food characteristics. Food during this time may be been drier. Finally, the drive frequency increased again for several generations, and the population was eventually eliminated in generation 14. Note that even though drive carrier frequency was below 1, nonfunctional resistance alleles likely rendered many females sterile in the last few generations. Successful population elimination was likely due to reduced embryo resistance, but fitness costs were also perhaps different than in individual cross-experiments. Two additional cages using either mostly older (and thus drier) food or only fresh food (cages 3 and 4, respectively) gave more consistent results, the former

reaching an equilibrium and the latter quickly increasing and eliminating the population.

Three cage trials were also conducted with the "site C" Cas9 element with the *CG4415* promoter (Fig. 7, Source Data—Data Set S6). In cage 5, the drive was still able to increase in frequency and eliminate the population, possibly due to the lower female heterozygote fitness cost with this Cas9 element (Fig. 6). However, the drive had an unexplained fitness advantage for the first couple generations in this cage, despite all generation 0 individuals have mothers of the same genotype and from the same batch. In another two replicate cages, the drive was also successful in eliminating the population, though the population size in these cages remained low.

To assess drive performance parameters based on cage data, a maximum likelihood method was applied, similar to previous studies[15,16,25,43]. We used a simple model with one gRNA and no functional resistance, the latter of which is likely a valid assumption due to four gRNAs[15]. Assuming only one gRNA may slightly underestimate drive performance[10] because drive/resistance allele heterozygous males can still do drive conversion if some gRNA cut sites remain wild-type. Females and males were assumed to have a different drive conversion efficiency based on drive inheritance data (Source Data—Data Set S5), though performance in cage Cas9 homozygotes may be slightly different than in individual crosses where flies only had one copy of Cas9. Embryo resistance was set at 5% for both drives. Varying this parameter between 0% and 10% had little effect on the results. The fitness of female drive heterozygous was allowed to vary and was inferred by the model.

We reasoned that large population size changes in this suppression drive, especially near the end, could be negatively affecting a model with a fixed effective population size. We, therefore, allowed it to vary, assuming that it was a fixed percentage of the average between the two generations for each generation transition. This produced fractional effective population sizes broadly consistent with previous results[15,16,25,43] (Table S3). The *rcd-1r* cage and the *CG4415* generations with drier food had low inferred fitness values, 0.29 and 0.36, respectively, accounting for the failure to eliminate two of the cages. For the successful *CG4415* reverse orientation cages with normal food, we inferred a fitness of 0.66, with the high end of the 95% confidence interval reaching 0.91, indicating a small to moderate fitness cost. For the *CG4415* site C cages, the lower population sizes and usually high drive fitness effect in cage 5 somewhat obscured results. The inferred fitness was 1.48, but the 95% confidence interval still reached well below 1. Further, Cas9 was homozygous in this experiment, which may have been particularly helpful to this construct with its lower germline cut rates. Combined with the benefit from multiple gRNAs compared to the model, this could indicate that the high inferred fitness was due to lower drive performance parameters in the model compared to the actual cage.

Overall, the fitness cost in the *CG4415* cages appeared to be substantially less than *nanos* or *rcd-1r*, and coupled with the greatly reduced rate of embryo resistance compared to *nanos* (which was likely over 50% for this target site[15]), this was sufficient to allow suppression of large, robust, *Drosophila* cage populations.

## Discussion

In this study, we compared several Cas9 promoters in CRISPR homing gene drives. The previously well-characterized *nanos* promoter had high germline cut rates and undetectable somatic expression but suffered from high embryo resistance allele formation due to maternally deposited Cas9. Among the ten other promoters we tested, *β2-tubulin* and *CG17658* had weak activity. This is somewhat unexpected for *β2-tubulin* because it can yield a strong sex bias from X-shredding[42], which is thought to require relatively high cut rates to support multiple-cutting. *zpg*, *CG7878*, and *CG3223* supported good drive conversion rates but, like *vasa*, had high somatic expression and

**Table 1 | Most impactful consequences of imperfect drive performance**

| Problem | Haplolethal modification | Haplosufficient modification | Female-fertility suppression |
|---|---|---|---|
| Low drive conversion | Slower drive | Slower drive | Much lower power |
| High germline resistance | Faster drive | Faster drive | Slower drive |
| High embryo resistance | Possible failure | Slower drive | Lower power[a] |
| High somatic expression | Fitness cost possible failure | Possible small fitness cost | Large fitness cost lower power[a] |

[a]Only if drive conversion is not very high.

embryo resistance allele formation rates. *rcd-1r*, *CG4415*, *shu*, and *mei-W68* could often achieve similar drive inheritance rates but substantially lower embryo resistance rates compared to *nanos*. However, *shu* had trouble maintaining high drive inheritance in systems with less active gRNAs, and *mei-W68* still had moderate embryo resistance. All of these suffered from somatic expression, but this was limited enough in *CG4415* that it was suitable for use in a suppression drive, resulting in successful population elimination.

To assess drive performance, visual markers allow genotyping based on phenotypes, avoiding laborious sequencing. Our EGFP target site drives and *yellow* split drives are both suitable for this purpose, and each has its advantages. Neither affects fly viability, but only EGFP allows assessment of male drive performance. Split Cas9 lines can be used more flexibly with other drive lines, and *yellow* allows easier assessment of female germline performance and somatic expression. It is also potentially more representative of somatic expression in a wider array of tissues. However, we saw similar somatic patterns in both of these drives. While our promoter choices were chosen for their strong germline expression and low expression in other tissues, they still have some non-germline expression[44], which can be in somatic cells or lead to embryo resistance. It is unclear if this native expression level would, in fact, be enough to produce the cleavage patterns we observed or if our Cas9 elements had different expression patterns due to missing regulatory sequences or other factors.

Genomic location certainly has a substantial effect on expression, as indicated by our tests with an alternate genomic site for the *CG4415* promoter split Cas9 element, which had lower somatic expression. Germline performance was also slightly worse, so it is unclear if tissue-specific expression was changed or if the expression was just generally lower. Indeed, while we generally desire high germline cut rates, there may be a considerable incentive to cease further increases in Cas9 expression once this has been achieved (aiming for the minimum level of Cas9 expression needed to achieve high drive conversion rates). Insertion of the PEST sequence at the C-terminus of Cas9 with the *nanos* promoter decreased embryo resistance rates somewhat in the EGFP target sites drive without affecting germline performance. However, in the *yellow* system with other promoters, the addition of a PEST sequence severely reduced Cas9 germline drive conversion activity. This is potentially also supported by our observation in this study and previous ones[7,15,16,20] that drive conversion was usually higher in males than in females for the *RpL35A* and *yellow-G* drives but usually higher in females than males in EGFP target drives or a drive targeting *cinnabar*. Embryo resistance, on the other hand, was higher in the EGFP and *cinnabar* drives. If embryo resistance is closely correlated with germline expression, then it is possible that generally higher Cas9 expression in males can explain these results. When expression is low, embryo resistance is low, and females may not have high germline cutting, leading to the persistence of many wild-type alleles and reduced drive conversion. Males with higher expression may still achieve higher drive conversion rates. However, as expression increases in both sexes, females now have higher drive conversion, while male drive conversion is actually reduced. With males now having more than sufficient germline expression, cleavage would tend to occur earlier on average. A similar pattern appears to be present in

mouse homing drives[14]. Resistance alleles are known to form in pregonial germline cells[7], and cleavage at this temporal phase may tend to produce more resistance alleles compared to drive conversion than later cleavage in or closer to the gametocyte stage. However, additional data would be needed for such a hypothesis to be strongly supported.

We also assessed alternate 3′ UTRs, albeit less systematically than promoters. It is possible that the 3′ UTR may interact with other regulatory elements at the mRNA stage, and it could also influence the rate of mRNA degradation. However, we generally found that the *nanos* 3′ UTR reduces embryo resistance and somatic expression compared to 3′ UTR elements matching the promoter, at least for a few of our promoters that already had low embryo resistance.

Haplolethal rescue drives have substantial advantages over ones targeting a haplosufficient gene. Resistance alleles are more easily eliminated in haplolethal drive system[16], and the drive can reach 100% final frequency even if it has fitness costs. If fitness costs are present in a haplosufficient rescue drive, the drive carrier frequency will be 100%, but the total drive allele frequency will be less, similar to CRISPR toxin-antidote drives[45–47]. However, embryo resistance can remove haplolethal rescue drive alleles, and somatic expression can form nonfunctional resistance alleles, leading to nonviability or heavy fitness costs (Table 1). In our haplolethal drive system, despite lower embryo cut rates than in other systems[16], promoters with high embryo resistance prevented successful egg production by females. However, we found several promoters with lower embryo resistance that appeared to also have no detectable negative effects from somatic expression. While somatic expression can certainly lead to heavy fitness costs in haplolethal drive systems[48], in this case, we tested them in homing drive systems, where drive conversion is possible in somatic cells[6]. Drive conversion would provide a second copy of the rescue gene, resulting in healthy cells. This, combined with the naturally low cut rates of this drive, likely allowed them to avoid detectable fitness costs, which is quite promising for future use of haplolethal homing drives in other species.

Though less problematic than functional resistance alleles, nonfunctional resistance alleles are more difficult to address and a primary obstacle for creating good gene drive systems, especially in suppression drives. All homing suppression drives targeting female fertility thus far have suffered from fitness costs in driving heterozygous females. This is critically important[49] because it reduces the genetic load (suppression power) of the drive when drive conversion is not close to 100%[50], which can result in the persistence of the population (Table 1). Even with high drive conversion, it can complicate suppression in spatial environments[51]. Embryo resistance has a similar effect, sterilizing daughters of female drive heterozygotes. Low drive conversion also reduces suppressive power, though this can be compensated by high total germline cut rates in new designs[52]. There is thus a higher incentive to develop improved promoters for suppression drives than otherwise well-designed modification drives. Our discovery of Cas9 promoters in *Drosophila* that minimize embryo resistance is thus encouraging for other species, considering that previously analyzed promoters in *D. melanogaster* tended to have either low germline activity or high maternal deposition.

Previously, our 4-gRNA drive targeting *yellow-G* failed to suppress a cage when paired with the *nanos* promoter[15], but with the *CG4415* promoter driving Cas9, suppression was successful. The main advantage of *CG4415* was the far lower rate of embryo resistance, but it appeared to have fewer fitness costs in heterozygotes in the cage study than *nanos* as well, unlike our test with *rcd-1r*, which performed worse than *nanos* despite also reducing embryo resistance. This is also somewhat contradictory to our egg viability experiments, though the 95% confidence intervals of fitness overlap, and different conditions in the cage experiments may result in different actual fitness costs compared to individual crosses, particularly since *yellow-G* is needed for egg shells, meaning that environmental conditions may strongly affect fitness. The observation of lower fitness costs with *CG4415* compared to *nanos* is also unexpected, considering the lower somatic expression in *nanos*. However, *yellow-G* is expressed in the ovaries, even if perhaps not in gametes. It is possible that even though *nanos* have lower general somatic expression, *yellow-G* was still disrupted in some ovary cells where it was needed, while the lower germline expression of *CG4415* (again, based on embryo resistance) caused less disruption to these cells and thus less fitness cost, allowing high efficiency and rapid success in the cage population.

While the performance of our promoters has revealed useful general information in the model organism *D. melanogaster*, they could potentially be applied to other species as well. This certainly seems to be the case with U6 promoters, which have been used to express gRNAs in every CRISPR gene drive study thus far. However, these have a less complex required expression pattern than Cas9, where we often desire restriction of cleavage activity to the germline rather than just accepting high expression everywhere. Recently, a drive system targeting *doublesex* was tested in the major crop pest *D. suzukii* and yielded good results with the *nanos* promoter for Cas9[53]. Performance was actually better than in *D. melanogaster* for drive conversion, though this could have been caused by the addition of a second nuclear localization signal. It is possible that other promoters would have similar performance in this and other closely related species, which could include important pests such as the medfly. However, in more distantly related species, such as mosquitoes, the situation is different. In an *Anopheles* homing suppression drive[11], the *zpg* promoter had low embryo resistance (likely under 10%[51]) and modest somatic activity, while in this study, both were high. *nanos* also had higher drive conversion than in *D. melanogaster* and had much lower embryo resistance[33]. *vasa* had high embryo resistance and somatic expression in both species[32]. The *shu* promoter in *Aedes aegypti* could support very high cut rates and drive conversion in the germline in some lines, while most other promoters failed to achieve this[18,28,29]. This contrasts with our results, where *shu* was a weaker promoter, achieving high efficiency only in the EGFP target line. All these comparisons, and our promoter assessment in general, have the important caveat of the exact length of promoter elements utilized. In some of our promoters, in particular, we sought to use shorter elements to avoid the coding sequence of other genes in an attempt to find more compact regulatory elements and avoid fitness costs from undesired transcription in different directions. However, even though specific promoters may have different expression patterns between species, several of the promoters we introduce here could at least be considered good starting points for trials in non-model species, particularly those more closely related to *D. melanogaster*.

Overall, our study demonstrated that homing drives could achieve high efficiency with several germline promoters, 5' UTRs, and 3'UTR regulatory elements for Cas9. In multiple systems, we identified the strengths and weaknesses of promoters and how they interact with varying drive elements. These regulatory elements could offer large advantages for drive systems in *Drosophila*, and their homologs could be useful as potential candidates in other species.

## Methods

### Ethical statement

This research complies with all relevant ethical regulations and was approved by the Peking University biosafety office.

### Plasmid construction

For plasmid cloning, reagents for restriction digest, PCR, and Gibson assembly were obtained from New England Biolabs; oligonucleotides from BGI and Integrated DNA Technologies; competent DH5α *Escherichia coli* from TIANGEN and New England Biolabs; and the ZymoPure Midiprep kit from Zymo Research. Plasmid construction was confirmed by Sanger sequencing. We provide annotated sequences of the final insertion plasmids and target genomic regions in ApE format[54] at GitHub (https://github.com/jchamper/ChamperLab/tree/main/Cas9-Promoters-Homing-Drive) (doi version: https://doi.org/10.5281/zenodo.10649892).

### Generation of transgenic lines

Embryo injections were conducted by Rainbow Transgenic Flies or Fungene. Donor plasmids (Table S1) were injected into *w[1118]* flies (500 ng/µL) together with a gRNA helper plasmid BHDabg1[20] (100 ng/µL) and TTChsp70c9[48] (450 ng/µL), which was used as the source of Cas9 for transformation. To expand populations, injected individuals were first crossed with *w[1118]* flies, with four females and two males in each vial. Their offspring with EGFP or DsRed fluorescence in the eyes, which usually indicated successful insertion of the transgenic cassette, were then crossed for several generations to obtain homozygotes. Adults expressing slightly brighter eyes were more likely to be homozygous.

### Fly rearing and phenotypes

All flies were cultured with Cornell standard cornmeal medium or with a modified version using 10 g agar instead of 8 g, the addition of 5 g soy flour, and without the phosphoric acid. Vials and cages were housed in a 25 °C incubator with a 14/10-h day/night cycle. Flies were anesthetized with $CO_2$ and screened for fluorescence using NIGHTSEA adapters SFA-GR for DsRed and SFA-RB-GO for EGFP. Fluorescent proteins were driven by the 3xP3 promoter for expression and visualization in the white eyes of *w[1118]* flies. DsRed was used to indicate the presence of the split drive allele or a synthetic target drive, and EGFP was used to indicate the presence of the Cas9 allele or served directly as the synthetic target. In split *yellow* drive systems, males usually only show natural color or yellow body color for both body and wings. However, females were considered as 'mosaic' if their body dorsal stripes or wing color were mixed yellow and natural. Each individual could also have one or both fluorescence colors indicating the presence of drive (DsRed) or Cas9/functional target (both EGFP).

### Cage study

For the cage study, flies were housed in 25 × 25 × 25 cm mesh enclosures. A line that was heterozygous for the split homing suppression drive allele[15] and homozygous for the supporting Cas9 allele was generated by crossing drive males to individuals with the Cas9 line for several generations. In each cross, we attempted to obtain homozygotes by selecting flies with brighter green fluorescence, and we eventually confirmed that the line was homozygous for Cas9 by PCR.

Males from this line (heterozygous for the split homing suppression drive and homozygous for Cas9) were crossed to Cas9 homozygotes, and similarly aged Cas9 homozygotes were also crossed to Cas9 homozygotes males in separate vials for two days. All males were then removed, and females were then evenly mixed and allowed to lay eggs in eight food bottles for two days. Bottles were then placed in cages, and 11 days later, they were replaced in the cage with fresh food. Bottles were removed from the cages the following day, meaning that future larger generations only laid eggs for one day per generation.

The flies were then frozen for later phenotyping for adult numbers and fluorescence. The egg-containing bottles were returned to the cage. This 12-day cycle with nonoverlapping generations was repeated for each generation.

Flies were occasionally given an extra day to develop if the bottles were due for replacement before approximately half of the pupae had visibly eclosed. Usually, most pupae would eclose after one day of egg laying followed by 11 days of development. When the population was observed to fall down to low levels near the end of successful cages, the flies were given fewer food bottles in which to lay eggs. The number was set to still keep a substantially lower relative density compared to the normal equilibrium population, and this had the effect of increasing the survival of larvae by reducing bacteria growth in bottles compared to the potential situation at near-zero density. This created a more robust population at a lower population density, reducing the Allee effect.

### Phenotype data analysis

Data were pooled from different individual crosses in order to calculate drive inheritance, drive conversion, germline resistance, embryo resistance, and other parameters. However, this pooling approach does not take potential batch effects into account (each vial is considered to be a separate batch, usually with different parameters, but sometimes with the same parent for egg count data, see Source Data), which could bias rate and error estimates. To account for such batch effects, we conducted an alternate analysis as in previous studies[10,15,16,25,47]. Briefly, we fit a generalized linear mixed-effects model with a binomial distribution (maximum likelihood, Adaptive Gauss-Hermite Quadrature, nAGQ = 25). This allows for variance between batches, usually resulting in slightly different parameter estimates and increased standard error estimates. This analysis was performed with R (3.6.1) and supported by packages lme4 (1.1-21, https://cran.r-project.org/web/packages/lme4/index.html) and emmeans (1.4.2, https://cran.r-project.org/web/packages/emmeans/index.html). The code is available on Github (https://github.com/jchamper/ChamperLab/tree/main/Cas9-Promoters-Homing-Drive). The alternate rate estimates and errors were similar to the pooled analysis (see Source Data).

### Genotyping

For genotyping, flies were frozen, and DNA was extracted by grinding flies from SNc9XSGr1 and SNc9XSGr2 lines separately in 200 μL DNAzol (Thermo Fisher) and an appropriate amount of 75% ethanol solution. The DNA was used as a template for PCR using Q5 Hot Start DNA Polymerase from New England Biolabs according to the manufacturer's protocol. The region of interest containing the promoter and 5' UTR fragment was amplified using DNA oligo primers Auto-B_left_S_F and Cas9_S1_R (see ApE file for primer sequences). This would allow amplification of the DNA fragment with a 30 s PCR extension time. After DNA fragments were isolated by gel electrophoresis, sequences were obtained by Sanger sequencing and analyzed with ApE software[54].

### Fitness cost inference framework

To quantify drive fitness costs, we modified our maximum likelihood inference framework[43]. Similar to a previous study[15], we analyzed our homing suppression drive targeting female fertility. The maximum likelihood inference method is implemented in R (v. 4.0.3)[55] and is available on GitHub (https://github.com/jchamper/ChamperLab/tree/main/Cas9-Promoters-Homing-Drive).

In this model, we make the simplifying assumption of a single gRNA at the drive allele site. Each female randomly selects a mate, and the number of offspring generated is reduced in drive/wild-type females if they have a fecundity fitness cost. No offspring are generated if females lack any wild-type allele. In the germline, wild-type alleles in drive/wild-type heterozygotes can potentially be converted to either drive or resistance alleles, which are then inherited by offspring. The genotypes of offspring can be altered if they have a drive-carrying mother and if any wild-type alleles are present. These alleles then are converted to resistance alleles at the embryo stage with a probability equal to the embryo resistance allele formation rate.

### Reporting summary

Further information on research design is available in the Nature Portfolio Reporting Summary linked to this article.

## Data availability

All raw data is available in the Source Data file. Plasmid sequences are available on GitHub (https://github.com/jchamper/ChamperLab/tree/main/Cas9-Promoters-Homing-Drive) version: https://doi.org/10.5281/zenodo.10649892. Source data are provided with this paper.

## Code availability

All code is available on GitHub (https://github.com/jchamper/ChamperLab/tree/main/Cas9-Promoters-Homing-Drive) (doi version: https://doi.org/10.5281/zenodo.10649892).

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

## Acknowledgements

This study was supported by laboratory startup funds from Peking University, the Center for Life Sciences, the National Natural Science Foundation of China (32270672 and 32302455), the SLS-Qidong Innovation Fund, and the National Institutes of Health award F32AI138476 to J.C.

## Author contributions

J.D., X.J., X.X., E.Y., R.Z., Y.Z., M.M., and J.C. carried out experiments. W.C. completed the maximum likelihood analysis. P.W.M. and J.C. provided supervision. J.D. and J.C. wrote the initial draft, and all authors provided editing and revisions.

## Competing interests

The authors declare no competing interests.
