## [Peer Review File · Nature Communications]

Reviewers' Comments:

Reviewer #1:

Remarks to the Author:

This ms describes the exploration of several *Drosophila* promoter fragments/putative promoters for the purpose of driving expression of Cas9 in homing-based gene drive systems, also some less comprehensive testing of 3'UTRs and a potential destabilisation of Cas9 protein. This is all in *Drosophila melanogaster*. One could question the point at which such work needs to switch from a model organism such as *D. melanogaster* to one in which such gene drives might potentially be used. To my mind, the various caveats that the authors correctly present in the Discussion indicate that this study is close to or beyond that border – several of the promoters used appear to have rather different properties in mosquitoes, for example. As the authors note, this is not necessarily due to species differences as it could relate to different choices of promoter fragments, for example – but both indicate, at least to this reviewer, a diminishing return from such studies in *D. melanogaster*.

This study has a number of limitations in common with other similar studies. "expression" and similar terms are used, but really the only metric is visible phenotype, there is no detailed analysis of actual expression, either overall or (e.g.) mRNA stability for studies on 3'UTRs, or protein stability (or even level) for studies involving putative destabilization domain (PEST). To be fair, I'm not sure how useful such studies would be, but the potential for more precise analysis is one of the attractions of model systems, without it another aspect of the argument for conducting these studies in a model falls away.

The corresponding author is a relatively new PI with an excellent record of innovation and analysis in this field. As one would expect, this paper is thoughtful and generally solid. However, I'm not sure this ms has the level of innovation and value of many of his previous papers, in particular in relation to the generalisability of the results to other systems. While certainly publishable (after addressing the issues below), I wonder whether it has broad enough interest for Nature Communications.

Major issues

Cage trials: Unless I'm missing something, the cage trial data represent single cages for each drive. I find that a little hard to believe, so perhaps I am indeed missing something. It's just impossible to separate out stochastic effects/unknown variation from "signal" in such a situation, and indeed the authors invoke such effects to explain some aspects. I'd expect duplicate (plus controls) in mosquitoes – and some reviewers clearly expect more – and perhaps triplicate in the rather easier *Drosophila* system. I appreciate this may not give new information, but it would give much more confidence in the conclusions currently being drawn. I really do not think this is publishable with single trials, so I hope I am indeed missing something.

Figs 3 and 4 – I'd expect some indication of uncertainty in these proportions, ie statistical analysis. Similarly Fig 9 (if more than one rep), S3 and elsewhere

Minor issues (in order of occurrence)

Abstract

"However, achieving successful outcomes with these drives often requires high performance" These are modelling results, as there are very few field data (natural systems and Wolbachia). This should be made clear, e.g. "Modelling indicates that..."

Introduction

Maybe cite <https://www.pnas.org/doi/10.1073/pnas.2020417117> after first use of "gene drive" to minimise need to say what it is (which the current ms does not really say at all, just that it is desirable!)

"specie" is coinage or currency, not the singular of species (which is "species", therefore "species' genome")

"When the drive has a higher fitness cost than functional resistance alleles, the drive allele frequency will TEND TO be reduced over time"

"an *A. gambiae*" rather than "a *A. gambiae*"

"Suppression drives typically target essential but haplosufficient genes without providing a rescue." Should cite Burt 2003 or a related subsequent review, since that's the origin of this idea

"Additionally, somatic Cas9 expression rendered female drive heterozygotes mostly sterile" strictly I think "Cas9 + sgRNAs". I do not think that Cas9 on its own causes such sterility. Similarly for

similar statements elsewhere in the ms.

"but none was able to avoid somatic expression like the nanos promoter" perhaps "but, unlike the nanos promoter, none was able to avoid..."

Methods

Plasmid construction: Good to provide plasmid sequences, but these need to be in a permanent repository with accession numbers rather than a (I think potentially transient) lab database, albeit an externally hosted one.

Generation of transgenic lines: "a gRNA helper plasmid" and "TTChsp70c9" need more description. Presumably they are in the set of plasmids referred to in "Plasmid construction", but "a gRNA helper plasmid" is hardly a precise reference. Probably need a table of plasmids, description and accession numbers, maybe as Supplementary. Possibly this info is in Table S1, described as "list of all constructs used in this study", but I could not see it there

Table S2 and main text – referring to distances relative to transcription start (5'UTR etc) is reasonable in summary, but sometimes the annotated transcription start changes. This table is OK so long as the complete sequences (which should then unambiguously describe the construct) are deposited as above.

Figure 3 "drive inheritance... WAS measured" not "are measured", similarly change to past tense elsewhere for things that were measured, calculated etc in the course of this work.

Though I think the comparison is probably OK, ideally the nanos data would have been (re-)measured alongside these other drives, rather than taken from a much earlier study.

Figs 3 and 4 – I'd expect some indication of uncertainty in these proportions, ie statistical analysis. Similarly S3 and elsewhere

"indicating the these promoters could affect each other" – that only strictly shows effects on the EGFP promoter. It would be quite interesting to know whether 3xP3 – which is widely used - did have the hypothesised effect (increasing somatic expression by affecting expression of an otherwise germline-specific promoter, but I can't see that clearly here.

"two sublines from with the CG4415 promoter" delete "from", I think

"had reversed orientation between the Cas9 promoter and 3xP3 (Figure S2A)". Fig S2 does not appear to have a panel A

"However, only two Cas9 elements driven by the nanos and CG7878 promoters had drive inheritance rates for females of over 70%. This is likely the result of reduced germline expression with these promoters, at least in females" Why "reduced germline expression" – weren't these promoters used before, with good results? I thought that was the point.

"when drive conversion in not nearly 100%" should be "is" not "in"

"we generally found that the nanos 3' UTR reduces embryo resistance and somatic expression compared to 3' UTR elements matching the promoter, at least for a few of our promoters" I'd agree that the reduced somatic effects (phenotype is measured rather than expression levels) indicates lower expression, perhaps through reduced mRNA stability. This may also apply to embryos, but I think nanos mRNA is localised in embryos, specifically to the posterior pole. If so, is it possible that decreased (somatic) embryonic cutting is due to relocalisation of Cas9, rather than reduction, and so perhaps associated with increased embryonic cutting in the germline? If so, what effect would this have? A similar detrimental effect but delayed by a generation? I do not necessarily expect a detailed response!

Reviewer #2:

Remarks to the Author:

Review for: “New germline Cas9 promoters show improved performance for homing gene drive”

Authors: Du J, Chen W, Jia X, Xu X, Yang E, Zhou R, Zhang Y, Metzloff M, Messer PW, Champer J

Reviewed by: Michael Smanski, smanski@umn.edu

Summary:

The authors report the design and performance of an impressively large number of gene drives in *Drosophila melanogaster*. Drive designs include (i) conventional (i.e., ‘non-split’) CRISPR/Cas-based homing drives targeting a transgene, and (ii) split CRISPR/Cas-based drives that home a sgRNA construct into a haplosufficient gene target, or (iii) split CRISPR/Cas-based drives that home a sgRNA construct into haploinsufficient gene target while simultaneously carrying a functional replacement allele. These designs comprise both suppression drives and replacement drives. For each general drive design there are roughly a dozen specific genetic instantiations. Each design has different cis-regulatory elements to control Cas9 expression. The inheritance rate and resistance allele formation rate are measured in bi-directional crosses. Cas9 nuclease activity is indirectly measured in germ and somatic cells for each drive. Finally, two cage trials are reported, one of which demonstrates successful population suppression after 14 generations.

The highlight of this study is the large amount of data collected for so many different gene drive designs. This paper is a tour de force from that respect. The paper is technically sound, and the data support the conclusions. There are two major weaknesses that should be addressed by revision, and neither require more experimentation. These are (i) a clearer description of the significance of this study, and (ii) improvements in the organization of the writing and figures. Both are described in more detail below my signature.

Overall, I recommend for publication with revisions. This is an exciting paper and will be well-read and cited in the field of genetic biocontrol.

Best,

Michael Smanski
smanski@umn.edu

Major revisions

1. The significance of this study needs to be better articulated in a clear and concise manner. The current draft highlights several improvements in the key performance parameters of a gene drive in the model organism, *D. melanogaster*. Generally, model organisms are used in place of application-relevant organisms because (i) they are easier to work with, and (ii) the knowledge gained is relevant to the real-world application species. The general gene drive designs tested here have already been demonstrated in *D. melanogaster*. This work consists of iterative improvements to drive performance via tweaking the cis-regulatory elements. Do the authors expect this work to predict the iterative improvement of gene drives in those organisms? To this

end, I found the paragraph starting at line 789 somewhat confusing. It started by stating that the promoters could be used in other species, but then pointed out many examples where the data from *D. melanogaster* was not predictive of the same elements in other species.

2. I found the written and graphical communication in this study fairly hard to digest. I am knowledgeable, but not expert, in the current literature of gene drive design and performance. I expect the average audience of Nature Communications will have less knowledge than me. I found myself having to re-read many sections multiple times to grasp the key points that the authors were trying to articulate. It is difficult to provide constructive criticism because it occurred throughout the manuscript. However, I will attempt to provide general suggestions on how to improve with specific illustrations in the draft. These illustrations are not comprehensive. I encourage the authors to look for ways to make the writing clearer and more concise throughout.
 - a. Have better separation of background, results, and discussion. There were multiple instances where background was provided in the results section. For example, Lines 278-301 in the Results do not communicate any results, but more background. These paragraphs would fit better in the background section. There were multiple times where interpretations of the results and their relationship to other studies are stated in the Results section. For example, Lines 367-369 related to X-shredding. While some journals opt to have combined 'Results and Discussion' sections that encourage this blend of observations and interpretations, that is not how the current manuscript is organized. Particularly with a study as technically-dense as this, it would improve readability to be more disciplined about separating Results and Discussion.
 - b. Write with shorter sentences. This will increase the readability. For example, the sentence starting in line 399 has six clauses and one parenthetical. The next sentence has five clauses and another parenthetical. Both could be split up into multiple short sentences. I suggest removing parentheticals throughout the paper, and replacing them with sentences.
 - c. The Figures would be more readable if they were broken up into more focused comparisons. Figure 1 is pretty good as-is. Somewhere should be stated that only non-functional resistance alleles are portrayed. Functional resistance alleles would result in different consequences in the bottom right. Figure 2 is fairly straightforward, but it would be easy to illustrate the 'reverse orientation' designs mentioned later (line 474) in this figure. Figures 3-6 are in my opinion unnecessarily complex. My preference would be to pull each of the types of bars into a separate bar graph (e.g, a stacked bar graph just comparing the blue/yellow stacked bars for the 12 designs as Subpanel 3A, next to a bar graph comparing the red bars in Subpanel 3B, etc.). I was not able to follow the explanation for why some of the blue/yellow bars extend above 100%, perhaps an illustration could help explain the better. Results from statistical tests should be included in the bar graphs and described in the legend. Figure 4 appears to have been compiled incorrectly in the file for reviewers. It is cut-off of the page.

Minor revisions

1. Line 44: "target species's genome"

2. Line 59: This might be a slight misunderstanding of field-specific constraints around the limits of the term 'maternal deposition', but I would consider maternal deposition to include cytoplasmic provision of Cas9/sgRNA to embryos whether or not the gene encoding those components was passed on. If the gene is passed on, then the current sentence is incorrect (drive conversion could still occur). The current sentence is only true when the gene encoding those components is not passed on in the embryo that also has maternal deposition of Cas9/sgRNA.
3. The apostrophe (3') is used repeatedly in place of the prime symbol (3'), which can be found in Microsoft Word using the 'insert symbol' button in the header.
4. Line 157: "competent *Escherichia coli* DH5 α "
5. Line 160: I did not check the GitHub, but I recommend sharing files in genbank (.gb) format instead of 'ApE format', as this is more universally accepted by different software.
6. Line 164: delete = sign.
7. Line 307: 'Each of these sites is downstream of two genes on either side to...' on either side of what? Can you rephrase to clarify?
8. Line 333: 3'-UTR should be 3'-UTRs.
9. Line 351: what is a 'single drive/EGFP drive heterozygote'?
10. Line 357: The choice behind reporting inheritance vs standard parameters is still not clear to me. Can you elaborate a bit?
11. Line 376: How are expression levels known to be 'substantially varying'?
12. Line 465: is somatic expression more prevalent in the *yellow* split drive, or just easier to observe (because you are not limited to seeing it in the eyes)?
13. Line 508: change 'expression' to 'activity', because this is more precise based on what was directly observed.

Dear Editor,

Thank you for this thorough assessment of our study. We have made an effort to implement all the requested revisions. In particular, we have added several additional multi-month cage studies utilizing the *CG4415* promoter, accounting for the length of time required for our revisions.

We hope that you and the reviewers agree that we have addressed at least most of the concerns with the manuscript and that it is now substantially improved.

Sincerely,
Jie Du and Jackson Champer (on behalf of all authors)

In the following, editor and reviewer comments are in bold, and our responses are in plain text.

REVIEWER COMMENTS

Reviewer #1 (Remarks to the Author):

This ms describes the exploration of several *Drosophila* promoter fragments/putative promoters for the purpose of driving expression of Cas9 in homing-based gene drive systems, also some less comprehensive testing of 3'UTRs and a potential destabilisation of Cas9 protein. This is all in *Drosophila melanogaster*. One could question the point at which such work needs to switch from a model organism such as *D. melanogaster* to one in which such gene drives might potentially be used. To my mind, the various caveats that the authors correctly present in the Discussion indicate that this study is close to or beyond that border – several of the promoters used appear to have rather different properties in mosquitoes, for example. As the authors note, this is not necessarily due to species differences as it could relate to different choices of promoter fragments, for example – but both indicate, at least to this reviewer, a diminishing return from such studies in *D. melanogaster*.

This study has a number of limitations in common with other similar studies. “expression” and similar terms are used, but really the only metric is visible phenotype, there is no detailed analysis of actual expression, either overall or (e.g.) mRNA stability for studies on 3'UTRs, or protein stability (or even level) for studies involving putative destabilization domain (PEST). To be fair, I’m not sure how useful such studies would be, but the potential for more precise analysis is one of the attractions of model systems, without it another aspect of the argument for conducting these studies in a model falls away.

The corresponding author is a relatively new PI with an excellent record of innovation and analysis in this field. As one would expect, this paper is thoughtful and generally solid. However, I'm not sure this ms has the level of innovation and value of many of his previous papers, in particular in relation to the generalisability of the results to other systems. While certainly publishable (after addressing the issues below), I wonder whether it has broad enough interest for Nature Communications.

Thanks for these your thorough review. See detailed responses below. In general, we certainly hoped that we'd see closer correlations between promoters among different species. However, the fact that we did not perhaps emphasizes the importance of having multiple candidate genes for other species. By showing more germline promoter candidates in this study, we have at least expanded the range of reasonably reliable candidates, even if there is no guarantee that an individual candidate may functional as desired in other species. In the future, we certainly hope to learn more direct lessons in fruit flies that could be applied to non-model organisms, but for now, we hope that the breadth of this study should still be of wide interest, as well as the first homing suppression drive that showed success in flies.

Major issues

Cage trials: Unless I'm missing something, the cage trial data represent single cages for each drive. I find that a little hard to believe, so perhaps I am indeed missing something. It's just impossible to separate out stochastic effects/unknown variation from "signal" in such a situation, and indeed the authors invoke such effects to explain some aspects. I'd expect duplicate (plus controls) in mosquitoes – and some reviewers clearly expect more – and perhaps triplicate in the rather easier Drosophila system. I appreciate this may not give new information, but it would give much more confidence in the conclusions currently being drawn. I really do not think this is publishable with single trials, so I hope I am indeed missing something.

For the past several publications, our cage studies have had a lower number of replicates than is typically found in other gene drive studies. However, this has always been accepted by reviewers (sometimes after some convincing) because our cages are usually much larger than that undertaken by other laboratories in terms of population size. In many cases because of this, one of our cages has substantially more statistical power than even triplicate cages from other labs. Stochastic variation would thus be minimized compared to smaller cages, where more extreme outcomes may be seen. Of course, one can argue that replicates could still be useful, but one can similarly argue that each generation could also be thought of as a "replicate" (at least partially independent from previous generations).

That said, our newer cages, despite some efforts, tend to have lower population sizes due to some unknown different in food quality (though still usually higher than other studies). As noted in the manuscript, there was also some uncertainty over variation with food quality, which may have particularly affected the performance of our suppression drive. We therefore decided to conduct some additional cage experiments to clarify this, and also to investigate our "best" promoter

(CG4415 at “site C). These are now fully integrated into the manuscript. the site C promoter performs quite well, and the previously successful construct was shown to have performance that is affected by food conditions. We believe that these results should substantially strengthen the conclusions of our study.

Figs 3 and 4 – I’d expect some indication of uncertainty in these proportions, ie statistical analysis. Similarly Fig 9 (if more than one rep), S3 and elsewhere

In general, uncertainty for these is fairly small (a few percent). However, these figures were originally so crowded, which together with “stacked” values, made us hesitant to include uncertainty. However, with new more clear figures, we had added in error bars for SEM. All uncertainties from two analysis methods are also available in the supplemental section (simple binomial and an alternate analysis that takes batch effects into account).

Minor issues (in order of occurrence)

Abstract

“However, achieving successful outcomes with these drives often requires high performance” These are modelling results, as there are very few field data (natural systems and Wolbachia). This should be made clear, e.g. “Modelling indicates that...”

Experimental cage studies have also indicated this, but it is certainly not based on field trials. We have clarified this.

Introduction

Maybe cite <https://www.pnas.org/doi/10.1073/pnas.2020417117> after first use of “gene drive” to minimise need to say what it is (which the current ms does not really say at all, just that it is desirable!)

This citation seems a bit controversial among some members of the gene drive community, but we have clarified this and cited some additional review articles.

“specie” is coinage or currency, not the singular of species (which is “species”, therefore “species’ genome”

Fixed.

“When the drive has a higher fitness cost than functional resistance alleles, the drive allele frequency will TEND TO be reduced over time”

Fixed.

“an *A. gambiae*” rather than “a *A. gambiae*”

Fixed.

“Suppression drives typically target essential but haplosufficient genes without providing a rescue.” Should cite Burt 2003 or a related subsequent review, since that’s the origin of this idea

Thank you. We now cite this paper.

“Additionally, somatic Cas9 expression rendered female drive heterozygotes mostly sterile” strictly I think “Cas9 + sgRNAs”. I do not think that Cas9 on its own causes such sterility. Similarly for similar statements elsewhere in the ms.

Yes, this is shorthand, assuming that gRNAs are already expressed everywhere. It’s worth clarifying, though, and we have now done this for the introduction. This was already noted at the beginning of the results section.

“but none was able to avoid somatic expression like the nanos promoter” perhaps “but, unlike the nanos promoter, none was able to avoid...”

Thank you. We have amended this sentence.

Methods

Plasmid construction: Good to provide plasmid sequences, but these need to be in a permanent repository with accession numbers rather than a (I think potentially transient) lab database, albeit an externally hosted one.

Sure, we’ve made a version with a doi, making it permanent.
doi.org/10.5281/zenodo.10649892

Generation of transgenic lines: “a gRNA helper plasmid” and “TTChsp70c9” need more description. Presumably they are in the set of plasmids referred to in “Plasmid construction”, but “a gRNA helper plasmid” is hardly a precise reference. Probably need a table of plasmids, description and accession numbers, maybe as Supplementary. Possibly this info is in Table S1, described as “list of all constructs used in this study”, but I could not see it there

We now specify that the gRNA helper plasmid is BHDabg1 plasmid.

Table S2 and main text – referring to distances relative to transcription start (5’UTR etc) is reasonable in summary, but sometimes the annotated transcription start changes. This table is OK so long as the complete sequences (which should then unambiguously describe the construct) are deposited as above.

Our sequences are fully annotated, and now have a permanent doi.

Figure 3 “drive inheritance... WAS measured” not “are measured”, similarly change to past tense elsewhere for things that were measured, calculated etc in the course of this work. Though I think the comparison is probably OK, ideally the nanos data would have been (re-)measured alongside these other drives, rather than taken from a much earlier study.

Fixed.

Figs 3 and 4 – I’d expect some indication of uncertainty in these proportions, ie statistical analysis. Similarly S3 and elsewhere

In general, uncertainty for these is fairly small (a few percent). However, these figures are so crowded, which together with “stacked” values, would create a lot of clutter if we included uncertainty. All uncertainties from two analysis methods are available in the supplemental section (simple binomial and an alternate analysis that takes batch effects into account).

“indicating the these promoters could affect each other” – that only strictly shows effects on the EGFP promoter. It would be quite interesting to know whether 3xP3 – which is widely used - did have the hypothesised effect (increasing somatic expression by affecting expression of an otherwise germline-specific promoter, but I can’t see that clearly here.

Yes, we still believe that somatic expression was true here due to the EGFP mosaics, but this is certainly just evidence in the other directly. We have clarified this.

“two sublimes from with the CG4415 promoter” delete “from”, I think

Fixed.

“had reversed orientation between the Cas9 promoter and 3xP3 (Figure S2A)”. Fig S2 does not appear to have a panel A

Fixed.

“However, only two Cas9 elements driven by the nanos and CG7878 promoters had drive inheritance rates for females of over 70%. This is likely the result of reduced germline expression with these promoters, at least in females” Why “reduced germline expression” – weren’t these promoters used before, with good results? I thought that was the point.

These promoters may have worked well in previous results, but the haplolethal homing drive has naturally lower gRNA cut rates. They may thus have been insufficient in this new situation. We have now clarified this, and the text reads:

“This is likely the result of reduced germline expression with these promoters, at least in females, coupled with reduced gRNA activity in this drive element compared to the split drive targeting *yellow* (best seen by comparing embryo cut rates in these systems^{16,20}).”

“when drive conversion in not nearly 100%” should be “is” not “in”

Fixed.

“we generally found that the nanos 3’ UTR reduces embryo resistance and somatic expression compared to 3’ UTR elements matching the promoter, at least for a few of our promoters” I’d agree that the reduced somatic effects (phenotype is measured rather than expression levels) indicates lower expression, perhaps through reduced mRNA stability. This may also apply to embryos, but I think nanos mRNA is localised in embryos, specifically to the posterior pole. If so, is it possible that decreased (somatic) embryonic cutting is due to relocalisation of Cas9, rather than reduction, and so perhaps associated

with increased embryonic cutting in the germline? If so, what effect would this have? A similar detrimental effect but delayed by a generation? I do not necessarily expect a detailed response!

Our terminology may need some tightening here. In this study, “embryo” effect tend to be from maternal deposition (even if they effect somatic cells), while “somatic” effect refers to new expression in somatic cells.

nanos mRNA is indeed localized, but most “embryo cutting” takes place early on, before this happens. Later, this localization may effect “mosaic” cutting, which could potentially be apparent, maybe having different effects based on the drive target and where it is expressed compared to the localization pattern. Without this localization, it’s possible that we might see somewhat more widespread cutting and phenotypes from maternally deposited Cas9, because it would be spread out for longer.

Based on previous results, “mosaic” phenotypes in the yellow gene usually mean that germline drive conversion can proceed normally, though in some cases, germline drive conversion is blocked by resistance, as though an embryo resistance allele formed at the zygote stage.

We are less clear on how localization would affect newly expressed Cas9. This process may take place before new zygotic/somatic expression, or it could also contribute to reduced cutting in most tissues,

Reviewer #2 (Remarks to the Author):

The authors report the design and performance of an impressively large number of gene drives in *Drosophila melanogaster*. Drive designs include (i) conventional (i.e., ‘non-split’) CRISPR/Cas-based homing drives targeting a transgene, and (ii) split CRISPR/Cas-based drives that home a sgRNA construct into a haplosufficient gene target, or (iii) split CRISPR/Cas-based drives that home a sgRNA construct into haploinsufficient gene target while simultaneously carrying a functional replacement allele. These designs comprise both suppression drives and replacement drives. For each general drive design there are roughly a dozen specific genetic instantiations. Each design has different cis-regulatory elements to control Cas9 expression. The inheritance rate and resistance allele formation rate are measured in bidirectional crosses. Cas9 nuclease activity is indirectly measured in germ and somatic cells for each drive. Finally, two cage trials are reported, one of which demonstrates successful population suppression after 14 generations.

The highlight of this study is the large amount of data collected for so many different gene drive designs. This paper is a tour de force from that respect. The paper is technically sound, and the data support the conclusions. There are two major weaknesses that should be addressed by revision, and neither require more experimentation. These are (i) a clearer

description of the significance of this study, and (ii) improvements in the organization of the writing and figures. Both are described in more detail below my signature.

Overall, I recommend for publication with revisions. This is an exciting paper and will be well-read and cited in the field of genetic biocontrol.

**Best,
Michael Smanski
smanski@umn.edu**

Thank you for these kind comments. We agree that one of the main features of this study is the sheer number of transgenic elements (with even more combinations).

Major revisions

1. The significance of this study needs to be better articulated in a clear and concise manner. The current draft highlights several improvements in the key performance parameters of a gene drive in the model organism, *D. melanogaster*. Generally, model organisms are used in place of application-relevant organisms because (i) they are easier to work with, and (ii) the knowledge gained is relevant to the real-world application species. The general gene drive designs tested here have already been demonstrated in *D. melanogaster*. This work consists of iterative improvements to drive performance via tweaking the cis-regulatory elements. Do the authors expect this work to predict the iterative improvement of gene drives in those organisms? To this end, I found the paragraph starting at line 789 somewhat confusing. It started by stating that the promoters could be used in other species, but then pointed out many examples where the data from *D. melanogaster* was not predictive of the same elements in other species.

Our results on this are somewhat mixed. There are certainly differences between our study and mosquitoes, which means that our best promoters may not always be suitable. However, we believe that taken as a whole, many of the promoters in our study could still be good candidates for trials in other species (eg, we have hopefully expanded the pool of “starting promoters to test” when moving into a non-model organism for situations with less prior Cas9 promoter knowledge in that species). To better clarify this, we have added a summary sentence to the referenced discussion paragraph stating:

“However, even though specific promoters may have different expression patterns between species, several of the promoters we introduce here could at least be considered good starting points for trials in non-model species, particularly those more closely related to *D. melanogaster*.”

Perhaps more generally, our study also shows that even in species where high maternal deposition is common, a thorough search can reveal promoters that avoid this. We thus followed the suggested change and added the following sentence earlier in the discussion:

“Our discovery of Cas9 promoters in *Drosophila* that minimize embryo resistance is thus encouraging for other species, considering that previously analyzed promoters in *D. melanogaster* tended to have either low germline activity or high maternal deposition.”

2. I found the written and graphical communication in this study fairly hard to digest. I am knowledgeable, but not expert, in the current literature of gene drive design and performance. I expect the average audience of Nature Communications will have less knowledge than me. I found myself having to re-read many sections multiple times to grasp the key points that the authors were trying to articulate. It is difficult to provide constructive criticism because it occurred throughout the manuscript. However, I will attempt to provide general suggestions on how to improve with specific illustrations in the draft. These illustrations are not comprehensive. I encourage the authors to look for ways to make the writing clearer and more concise throughout.

We’ve gone through the manuscript again and made several small clarity improvements, in addition to those detailed below. That said, gene drive can be a complicated topic, and this manuscript deals with many detailed aspects of gene drive. We hope that these changes still have resulted in substantial improvements, even if some parts remain difficult.

a. Have better separation of background, results, and discussion. There were multiple instances where background was provided in the results section. For example, Lines 278-301 in the Results do not communicate any results, but more background. These paragraphs would fit better in the background section. There were multiple times where interpretations of the results and their relationship to other studies are stated in the Results section. For example, Lines 367-369 related to X-shredding. While some journals opt to have combined ‘Results and Discussion’ sections that encourage this blend of observations and interpretations, that is not how the current manuscript is organized. Particularly with a study as technically-dense as this, it would improve readability to be more disciplined about separating Results and Discussion.

In many papers, we have preferred to have the introduction and especially the discussion cover “high-level” concepts, putting some smaller things directly relevant to specific results in the results section. In this manuscript, this is perhaps even more important due to the already long length of the discussion. However, we likely went a bit too far with this. We have moved a few parts out of the results section, though we haven’t completely purified it. Together with the other changes below, these revisions should help the results section be substantially more understandable.

For the particular points mentioned, much of the first results subsection (including Figure 1) has been blended into the introduction, so the introduction size has only modestly increased due to removal of repetitive material. We also moved the X-shredding part to the beginning of the discussion and expanded this part of the discussion to include a summary of all the promoters we used. This should make the manuscript somewhat more accessible. A few other parts have also been moved from the results.

b. Write with shorter sentences. This will increase the readability. For example, the sentence starting in line 399 has six clauses and one parenthetical. The next sentence has five clauses and another parenthetical. Both could be split up into multiple short sentences. I suggest removing parentheticals throughout the paper, and replacing them with sentences.

This sentence was a particularly bad example and has been split into two. We've gone through the whole manuscript and made a large number of other changes to clarify more complex sentences. Most of these are in the results section, which will hopefully improve readability. Many parentheticals are removed when the flow of the writing can be continued with their information better incorporated, though a few remain. We've also split, clarified, or simplified several other overly long sentences.

c. The Figures would be more readable if they were broken up into more focused comparisons. Figure 1 is pretty good as-is. Somewhere should be stated that only nonfunctional resistance alleles are portrayed. Functional resistance alleles would result in different consequences in the bottom right. Figure 2 is fairly straightforward, but it would be easy to illustrate the 'reverse orientation' designs mentioned later (line 474) in this figure. Figures 3-6 are in my opinion unnecessarily complex. My preference would be to pull each of the types of bars into a separate bar graph (e.g, a stacked bar graph just comparing the blue/yellow stacked bars for the 12 designs as Subpanel 3A, next to a bar graph comparing the red bars in Subpanel 3B, etc.). I was not able to follow the explanation for why some of the blue/yellow bars extend above 100%, perhaps an illustration could help explain the better. Results from statistical tests should be included in the bar graphs and described in the legend. Figure 4 appears to have been compiled incorrectly in the file for reviewers. It is cut-off of the page.

Thank you for this comment. Perhaps it should have been clear to us before, but this was our wakeup call to make some improvements. We've made several adjustments.

Figure 1 now clearly specifies in the bottom-right portion (which is now clearly labeled as "B") that nonfunctional resistance alleles are indicated. In the rest of the figure, we want to keep it general for all resistance alleles.

For Figure 2, we could see including the other drives. However, the "reverse" orientation aspect of some drives didn't appear to be of primary importance, so we thought that the supplement was a better place for them.

For Figures 3-4 and S3, we agree that they are too busy. We think that in many cases, it's important to keep closely related bars together for comparison, but we went too far with this in the original version. We have split up each of these figures into two parts, one for germline performance and another for effects from maternal deposition and somatic expression. While Figure 4 is a little cramped, we believe that these figures are now much improved, and we thank the reviewer for bringing this to our attention. We had enough room to add "wild-type inheritance" (even though it was the complement of drive and resistance allele inheritance, visualizing it should be useful), and each figure part now only has 3-4 bars instead of 6-7.

For Figures 5-6, we could have split it up into drive inheritance and fitness cost, but after playing around with some versions, we didn't think it was helpful enough to justify. With only four bars per drive (and two easily distinguished classes of data), as well as a smaller number of drives, we thought the current versions still worked a bit better.

Some of the bars extended above 100% because each bar was derived from different subsets. For example, both-sex inheritance could be 90%, and resistance allele formation among male progeny could be 11%. This makes it above 100%, but really, perhaps drive inheritance among males was 89%, and drive inheritance among females was 91% (the difference just due to random chance). Thus, while perfectly accurate estimates would add up to 100%, uncertainty between different measurements could lead to something else. Rather than dwell on this, we now separate the bars in the “germline performance” part of the new figures. This sort of “sweeps the issue under the rug”, but by adding “wild-type” inheritance, there is now a much easier way to see how many alleles were uncut or became functional resistance (previously, this was the reason that we stacked the bars). We removed all reference to bars going above 100%, because this is now a minor artifact that readers could learn among by checking the supplemental data (though we still specify which data is from male progeny and so forth). This is another advantage of simplifying the figures.

In this new revision, we have added error bars to the SEM in all the figures, now that they are less cluttered (two methods of calculating uncertainty are also in the supplemental spreadsheets). There is some argument toward creating a full range of statistics comparing every drive, but this would require a full table, and at any rate, we didn't want to focus on drives with small differences (the text is already fairly long) where statistical comparisons would be needed due to inability to distinguish significance at a glance.

Figure 4 has so many different Cas9 alleles that it is horizontally oriented, which was necessary to keep text size for the Cas9 elements legible. This may have caused formatting issues. Hopefully, it will be improved in the new version, but either way, we are including all the main figures as separate files in our revision that should be accessible separately.

Minor revisions

1. Line 44: “target species’s genome”

Fixed.

2. Line 59: This might be a slight misunderstanding of field-specific constraints around the limits of the term ‘maternal deposition’, but I would consider maternal deposition to include cytoplasmic provision of Cas9/sgRNA to embryos whether or not the gene encoding those components was passed on. If the gene is passed on, then the current sentence is incorrect (drive conversion could still occur). The current sentence is only true when the

gene encoding those components is not passed on in the embryo that also has maternal deposition of Cas9/sgRNA.

Yes, we fully agree. Maternal deposition occurs regardless of drive inheritance. Looking at this sentence, it might have been a bit too vague about what the possibilities actually are. Basically, it makes resistance alleles regardless of the presence of a drive allele, though drive conversion is only theoretically possible when a drive allele is present in the first place). However, perhaps we tried to be a bit too concise. The sentence was extended to further clarify the case.

3. The apostrophe (3') is used repeatedly in place of the prime symbol (3'), which can be found in Microsoft Word using the 'insert symbol' button in the header.

Surprising, we were not aware that these were structurally different... fixed in the new version.

4. Line 157: "competent Escherichia coli DH5 α "

Fixed.

5. Line 160: I did not check the GitHub, but I recommend sharing files in genbank (.gb) format instead of 'ApE format', as this is more universally accepted by different software.

We have added gb files to increase readability.

6. Line 164: delete = sign.

Fixed.

7. Line 307: 'Each of these sites is downstream of two genes on either side to...' on either side of what? Can you rephrase to clarify?

We have clarified this sentence.

8. Line 333: 3'-UTR should be 3'-UTRs.

Fixed.

9. Line 351: what is a 'single drive/EGFP drive heterozygote'?

"Single" here was just causing confusion. It refers to each vial having only one drive individual, but saying it is this way here was not necessary (this info is in the methods).

10. Line 357: The choice behind reporting inheritance vs standard parameters is still not clear to me. Can you elaborate a bit?

Sure. Consider the haplolethal-targeting drive. If it had a 0% drive conversion and 100% germline resistance allele formation, it would have a drive inheritance rate of 100% (because all individuals inheriting the resistance alleles are nonviable). Of course, if drive conversion was 100% and germline resistance allele formation 0%, it would also be 100%. This makes it impossible to know what the two parameters are based on just the drive inheritance rate. The

reason this happens is that offspring survival is different between drive and resistance allele carrying individuals.

For most drives, there is no difference, but for the haplolethal drive, it might have a big effect. We thus elected to use “inheritance” here to ease comparisons between the different drives. Reporting of inheritance is not uncommon, though, so we don’t think it should raise too many eyebrows. In the manuscript, we attempted to clarify this part.

11. Line 376: How are expression levels known to be ‘substantially varying’?

Good point. It’s not known for sure. We not directly clarify this as “based on highly variable embryo resistance allele formation rates, though this is an indirect proxy”.

12. Line 465: is somatic expression more prevalent in the yellow split drive, or just easier to observe (because you are not limited to seeing it in the eyes)?

Somatic Cas9 expression is easier to observe and distinguish in drive heterozygote crosses because somatic Cas9 expression only occurs in progeny with Cas9 allele, which is separated from the gRNA-containing drive allele in this system. More generally, somatic Cas9 expression could be observed from eye mosaic fluorescence with the EGFP target drives, but in the yellow system, it could be observed over the whole body, especially in the wings and abdomen dorsal stripes. Though we saw substantial differences in expression between different Cas9 elements, we don’t necessarily have any reason to conclude that somatic activity was enhanced overall in the yellow target system verses the EGFP targets.

13. Line 508: change ‘expression’ to ‘activity’, because this is more precise based on what was directly observed.

Thank you. We have amended it.

Reviewers' Comments:

Reviewer #1:

Remarks to the Author:

I think the authors have done an excellent job of revising the manuscript. I am not convinced by their arguments for non-replication of cage experiments, but that's a bit moot as they have done more cage experiments. Overall, an excellent paper. I remain dubious about the value of more *Drosophila* (*melanogaster*) experimentation, but the authors make their case well.

[On cages – really just to explain my position, not because it's all that relevant now – I'd agree that the larger cage sizes (broadly, a few thousand rather than a few hundred) individuals reduces stochastic effects in the sense of individual flies, low-ish frequency genotypes, etc. However, there are plenty of other issues that can influence the outcomes that can be controlled by replication. One obvious one is environmental effects, and indeed the authors present one aspect of this (food quality) as an explanation for some of their observations. Clearly, replication could control for this – though, to be fair, these need to be independent replicates; in many cases replicate cage experiments are conducted in parallel, with some shared environmental features and so not entirely independent, thereby reducing the relative value. Plenty of oddities in their Fig 7 data that one might wonder about – why do cage 3 and 5 populations crash, then recover, then crash again? Why does cage 3 crash at all when the drive does not take off? [all these comments assume I'm reading these graphs correctly, these are just illustrative points so I'm not being super-careful]. Replication might have helped with that! Or not. To be clear, I'm not enough of a statistician to be certain.]

Reviewer #2:

Remarks to the Author:

I have read the revised manuscript, and believe that the authors adequately addressed my main critiques of the first draft. I recommend this version for publication.